# Iron Oxide Nanoparticles: A Review on the Province of Its Compounds, Properties and Biological Applications

**DOI:** 10.3390/ma16010059

**Published:** 2022-12-21

**Authors:** Priyannth Ramasami Sundhar Baabu, Hariprasad Krishna Kumar, Manju Bhargavi Gumpu, Jayanth Babu K, Arockia Jayalatha Kulandaisamy, John Bosco Balaguru Rayappan

**Affiliations:** 1School of Chemical and Biotechnology, SASTRA Deemed University, Thanjavur 613 401, Tamil Nadu, India; 2School of Advanced Materials Science and Engineering, Sungkyunkwan University (SKKU), Suwon 16419, Republic of Korea; 3Acrophase, Indian Institute of Technology Madras, Chennai 600 036, Tamil Nadu, India; 4Department of Physics, National Institute of Technology, Tiruchirappalli 620 015, Tamil Nadu, India; 5Centre for Nanotechnology & Advanced Biomaterials (CeNTAB), SASTRA Deemed University, Thanjavur 613 401, Tamil Nadu, India; 6School of Electrical & Electronics Engineering, SASTRA Deemed University, Thanjavur 613 401, Tamil Nadu, India

**Keywords:** iron oxide polymorphs, nanoparticles, properties, bio-imaging, electrochemical biosensors, biomaterials

## Abstract

Materials science and technology, with the advent of nanotechnology, has brought about innumerable nanomaterials and multi-functional materials, with intriguing yet profound properties, into the scientific realm. Even a minor functionalization of a nanomaterial brings about vast changes in its properties that could be potentially utilized in various applications, particularly for biological applications, as one of the primary needs at present is for point-of-care devices that can provide swifter, accurate, reliable, and reproducible results for the detection of various physiological conditions, or as elements that could increase the resolution of current bio-imaging procedures. In this regard, iron oxide nanoparticles, a major class of metal oxide nanoparticles, have been sweepingly synthesized, characterized, and studied for their essential properties; there are 14 polymorphs that have been reported so far in the literature. With such a background, this review’s primary focus is the discussion of the different synthesis methods along with their structural, optical, magnetic, rheological and phase transformation properties. Subsequently, the review has been extrapolated to summarize the effective use of these nanoparticles as contrast agents in bio-imaging, therapeutic agents making use of its immune-toxicity and subsequent usage in hyperthermia for the treatment of cancer, electron transfer agents in copious electrochemical based enzymatic or non-enzymatic biosensors and bactericidal coatings over biomaterials to reduce the biofilm formation significantly.

## 1. Introduction 

Iron has always been the quintessential transition element that has been extensively observed, studied, and engineered. Its ability to be modeled into anything that can be put into use in daily life with desirable features has attracted material scientists around the world to explore the wholesome properties of iron. Iron oxides, the abundantly found forms of iron on Earth, are the most preferred for synthesizing multi-functional magnetic nanomaterials and has applications as contrasting agents in MRI [1], magnetic storage device in computers [2] and efficient electrodes to produce hydrogen in solar cells via photo-assisted electrolysis of water [3]. Magnetite, Hematite, Iron oxide β-phase, Maghemite, Goethite, Akaganeite, Lepidocrocite, Feroxyhyte, Bernalite, Iron (II) hydroxide, Ferrihydrite, Wüstite, Schwertmannite are the known phases of iron oxides. These compounds are well characterized by their trivalent state, distinct colors, and low solubility [4]. A new stable polymorph, in addition to those mentioned above, has been found under the iron (III) family, which is known to be a ζ-Fe_2_O_3_ phase [5]. 

Hence, there is a need to discuss the different synthesis methods by which iron oxide nanoparticles are synthesized to benefit with the required surface, electrical, optical, and mainly magnetic properties for innumerable applications, as illustrated in Figure 1. In addition to the methods, insights on the structure of these nanoparticles are discussed in order to understand how different synthesis protocols can result in nanoparticles with different sizes, structures, and properties. 

## 2. Formation of Iron Oxide Nanoparticles 

Iron oxide nanoparticle synthesis remains challenging, as the properties that are expected out of these particles are highly dependent on the experimental procedure adopted. This is because the particle size, oxidation state, stoichiometry or the magnetic property must be “traded off” with any other feature, thereby losing one of the essential features. In addition, these nanoparticles are inherently colloidal, making it difficult to achieve small sized nanoparticles exhibiting a higher saturation magnetization co-efficient [6]. Hence, different methods have been concocted and optimized to develop multi-functional iron oxide nanoparticles. 

Some of the reported methods in the literature are hydrothermal [7,8,9,10,11], coprecipitation synthesis [12,13,14], microemulsions techniques [15,16], electrospray synthesis [17], sonochemical [18,19], electrochemical [20,21], microwave assisted synthesis [22,23,24], aerosol/vapor phase approach [25], microplasma under atmospheric pressure [26] and green synthesis [27,28,29,30,31]. The conventional and widely used synthesis methods are discussed in the subsequent sections, highlighting their advantages and disadvantages.

### 2.1. Coprecipitation Synthesis

Magnetic iron oxide nanoparticles are predominantly synthesized by a wet chemical coprecipitation method, which is more effective than the other methods available for synthesis. The mechanism involved is simple, involving the reaction of an aqueous solution of ferrous and ferric salts in the presence of an alkali furnished with enough incubation time for the precipitation of the magnetite, maghemite and ferrous hydroxide polymorphs of iron oxides. The synthesis method can be varied to realize the different phases of iron oxide by varying the pH of the solution. Magnetite is obtained with the precursor of basic nature (between 8–14) along with Fe^2+^/Fe^3+^ ratio equaling to 2:1 [6]. As magnetite is highly sensitive to the presence of oxygen, it is easily oxidized to maghemite. Thus, the synthesis of magnetite is preferred under non-oxidizing conditions. The reason for magnetite oxidation is that iron ions desorb from the magnetite surface, forming cationic vacancies in order to balance the structure’s charge, leading to the formation of maghemite. The distribution of Fe^2+^ and Fe^3+^ ions in the spinel structure is the key difference that discerns the magnetite and maghemite forms of iron oxide, where the latter is distinctive in its crystal structure with the presence of cationic vacancies in the octahedral sites. It is trivial in nanoscience that the employed synthesis technique has an influence over the ordering of the vacancies, resulting in symmetry lowering and the formation of superstructures [4,32]. 

The principal advantage of this method is its ability to scale up the yield. However, a major downside of this method is in obtaining a broad particle size distribution post-synthesis, as kinetic factors govern crystal growth. The reason for this is that nucleation sites and their growth by solute diffusion towards the crystal surface are deeply influenced by the pH, temperature, precursor salts used, reaction rate and the Fe^2+^/Fe^3+^ ratio [6]. Hence, the appropriate reaction conditions and precursors must be selected in order to obtain the desired shape-defined nanoparticles with the expected morphological characteristics. It was later reported, by Babes et al., that the Fe^2+^/Fe^3+^ ratio is pivotal to enhancing the size of the particles [33].

### 2.2. Hydrothermal Synthesis 

Hydrothermal synthesis involves the use of autoclaves maintained at high pressure and temperature conditions. The reactions take place inside a closed chamber, preferably an autoclave, where the aqueous iron precursor solutions form ferrites in two stages: hydrolysis, followed by mixed metal hydroxide oxidation. Although this method can be employed for other phases of iron, magnetite generation remains the most “profitable”. Larger sizes of magnetite were obtained when the content of water used in the reaction was higher, which means that the solvent takes up the governing role in deciding the particle size. Similar to the coprecipitation method, the particle size of the synthesized nanoparticles by this method is influenced and determined by the nucleation and growth rate. It is an established fact that if the nucleation phase is shorter than the growth phase at higher temperatures, the obtained nanoparticles would show a reduction in their sizes during the growth phase, notwithstanding grain growth, if the reaction time is extended [6,34]. 

Iron oxide nanoparticles with sizes 4–27 nm was synthesized easily using this method, with different iron precursors (iron pentacarbonyl and iron (III) acetylacetonate) and organic solvents and surfactants [6]. In this method, the morphological characteristics of the nanoparticles can be fine-tuned by optimizing process parameters such as reaction time, temperature, pressure, and solution parameters, for example, through the concentration of the reactants, solvent type, and type of complexing agents [35]. The hydrothermal method coupled with the microwave technique has been shown to efficiently produce superparamagnetic iron oxide nanoparticles (SPIONs) at an industrial scale [6].

### 2.3. Sol-Gel Synthesis 

The sol-gel technique is a two-step process involving the initial step of the hydroxylation of the precursor in a liquid state, followed by its condensation, from which a sol of nano-scale particles originates. A 3D metal oxide network, known as the wet gel, would be obtained upon the sustained condensation and inorganic polymerization processes. Additional heat treatments can be employed to acquire the final crystalline state. Reaction conditions, such as temperature, solvent, pH, agitation, and the nature of precursors taken, account for the structure and properties of the gel, which influences their reaction kinetics and growth. Maghemite nanoparticles, in the size range of 6–15 nm, have been easily obtained by heating the gels directly at 673 K. 

This method possesses some advantages, such as the fabrication of custom-made nanoparticles with a predetermined structure, the ability to obtain pure amorphous phases, particle size control and monodispersity of the solution, regulated microstructure and the homogeneity of the products obtained from the reaction, as well as the relatively low temperature conditions required for processing the materials. The sol-gel method has been extensively used to implant iron oxide nanoparticles onto suitable matrices to use them for different applications as multi-functional materials [6,34,35]. For instance, Solinas et al. [36], have incorporated maghemite nanoparticles in a silica matrix that is inert and temperature resistant. This is achieved by a high surface of evaporation to volume ratio selection in the gelation process, leading to the formation of small-sized iron oxide nanoparticles due to the high porosity of silica. If magnetite must be produced, a high surface of evaporation to volume ratio values and low temperature ought to be used.

### 2.4. Microemulsion Synthesis 

When an immiscible phase, present in the form of small droplets, is dispersed in a continuous phase, they give rise to transparent solutions called microemulsions. It is to be noted that the immiscible phase could be non-polar or polar. These microemulsions can either be oil-in-water or water-in-oil systems. The pivotal mechanism, here, is the reduction in the surface tension between the two phases to keep the droplets stable by decreasing the total surface energy of the solution system, which is achieved using surfactants. This leads to the formation of micelles, which itself results in the dispersion of the obtained iron oxide nanoparticles. The major advantage associated with this method is that various self-assembled structures, such as spherical, lamellar, and cylindrical and bio-continuous microemulsions, can be fabricated easily. However, whilst self-assembled structures can be formed, the aggregation of the prepared nanoparticles require repeated washing processes and stabilization treatments, despite the use of surfactants. Goethite nanorods using ferrite nitrate as the precursor in a high basic medium were obtained as single-crystalline structures, with a length of 106 nm and a diameter of 8.2 nm [6,35,37]. In addition, monodisperse maghemite nanoparticles were prepared with a size distribution of 3.5 nm and a high saturation magnetization [38]. The literature suggests that this technique is well suited for the preparation of nanoparticles less than or equal to 10 nm [15]. 

### 2.5. Sonochemical Synthesis 

The sonochemical method uses ultrasound to form new phases of the material and highly monodisperse nanoparticles, owing to the very high rate of cooling employed in the process. At high temperatures, the salt precursors are converted to nanoparticles because of the hot spot being formed due to the swift collapse of sonically induced cavities. This method is known to produce iron oxide nanoparticles with unique properties [6]. The study performed by Vijaykumar et al. [18] suggests that magnetite nanoparticles of 10 nm with a superparamagnetic property were synthesized purely using sonochemical means. The methodology was later improved by changing the surfactants and precursors used to enhance the magnetic properties of the obtained nanoparticles. However, whilst the sonochemical technique allows us to prepare mono disperse nanoparticles with various morphologies and narrow particle sizes, it fails as an industrial method for large-scale production [34,35].

### 2.6. Electrochemical Synthesis 

A very recent synthesis technique for the iron oxide nanoparticles, and particularly the magnetite and maghemite polymorphs, is the electrochemical method. Out of all the discussed methods, this method is the simplest and provides control of the particle size as extreme reactive conditions are not necessary, except that the temperature employed is not more than the electrolyte’s boiling point. As the current and potential are the changeable parameters in the electrochemical method, kinetic control and thermodynamic control can be established over the synthesis. Although the experiments are effortless, as the reactions happen at room temperature, the required nucleation sites and growth are not achieved, leading to randomly ordered nanoparticles or amorphous products [6,20]. There are other factors proposed by Cabrera et al. [21], who aimed to conducting a smooth electrochemical synthesis of iron oxide nanoparticles. The distance permeating between the anode and cathode is one such paramount factor, as it governs the diffusion kinetics of hydroxyl ions for its migration from the cathode (where it is formed) to the anode, forming iron hydroxide as the product. Upon increasing this distance, the required pH to produce iron hydroxide will not be reached due to the insufficient number of hydroxyl ions that would not have migrated to the anode. Hence, it was found that the optimum distance between the electrodes is 5 cm or less. The current density is another parameter to be considered as the magnetite precipitate amount increases with an increase in the current density. This is because a larger current density ensures the opening of more active sites to facilitate the reaction and form the precipitate [21]. 

### 2.7. Green Synthesis 

This synthesis method has gained significant interest among material scientists for synthesizing iron oxide nanoparticles, primarily because of its environmentally friendly nature, with the produced nanoparticles being less toxic compared with those synthesized by any of the above-mentioned methods. Various biological sources, such as plants [30], fungi [39], bacteria [40,41], viruses [28] and algae [42], have been extensively used for producing iron oxide nanoparticles in a range of shapes. In this regard, these particles are usually observed to have greater stability, owing to the presence of biomaterials playing the role of reducing, capping and stabilizing agents. Any form of green synthesis involves two main steps; namely, bioreduction and biosorption, wherein metal ions undergo a chemical reduction to their stable forms, in the former step, and the subsequent binding of these metal ions onto the surfaces of living organism is used to form stable complexes, take place in the latter step [28]. 

Lakshminarayanan et al. made use of the one-pot green synthesis method to synthesize hematite nanoparticles from the leaf extract of *Bauhinia tomentosa*, which was subsequently used to produce 1,3-diolein through enzyme-mediated processes [27]. Numerous other plant extracts from seeds, seed coat, flowers, peels, petals, fruits, and leaves from plants such as green tea, *Solanum trilobatum*, *Persia Americana*, *Camellia sinsensis*, were used to make iron oxide nanoparticles of varying sizes and polydispersity. The sheer diversity of options available for extracts render plants the most sought-after source for green synthesis of nanoparticles. Secondly, micro-organisms, such as fungi and bacteria, are considered as synthesis sources owing to their superior tolerant nature and bio-accumulation capability, due to their secretion of large amounts of extracellular hydrolyzing enzymes, such as reductases to reduce metals. In comparison with fungi, bacteria are more extensively used because of their faster doubling time and their ability to grow under extreme/adverse conditions [28]. 

## 3. Structure of Iron Oxide Nanoparticles 

Historically, the structure of nanoparticles has been the most important factor contributing to the function of a material that is synthesized by various methods. The different characterization techniques that were recently developed enable us to visualize the atomic realm with accuracy, thereby allowing us to engineer the properties of the material and, thus, synthesize them. In science, there has always been a debate on whether it is structure-based function or function-based structure, to which an answer has not yet been found due to convincing arguments on either side. Hence, the importance of structure can be realized for developing iron oxide nanoparticles with specialized functions. In this section, the morphological and crystal structures of different phases of iron oxides are discussed, highlighting their properties and the functions they elicit. The crystal structures of the various polymorphs of iron oxide are shown in Figure 2. 

### 3.1. Magnetite

The most abundant and important iron oxide, magnetite, contains both Fe^2+^ and Fe^3+^ in an equal stoichiometric distribution; however, this ratio is often vulnerable and results in a Fe^3+^ deficient layer. It is crystallographically characterized by an inverse spinel structure and a face-centered cubic unit cell with an edge length of 8.396 Å and each of the unit cells consist of 8 Fe^2+^, 16 Fe^3+^ and 32 O_2_ atoms, as shown in Figure 2a. As is the case of the inverse spinel structure of a mineral with general formula (A^2+^B_2_^3+^O_4_^2−^), and comparing it with Fe_3_O_4_ (magnetite), it is observed that Fe^2+^ and half of Fe^3+^ ions populate the octahedral sites, while the other half of the Fe^3+^ ions populate the tetrahedral sites. The reason for the divalent Fe ions occupying the octahedral sites is that they come into possession of higher crystal field stabilization energy when the d-orbitals of the iron atom degenerates in the presence of a ligand. On the other hand, the trivalent Fe ions neutralize their crystal field stabilization energy by filling two octahedral sites and a tetrahedral site. Octahedron and rhombodecahedron structures are the other crystal types of magnetite. These crystal structures are characteristic of magnetite because they display only one form of crystal structure, which is in its closed form, in contrast to hematite or goethite, as the {111} plane by which these structures are made encloses all of the space [4]. 

Magnetite nanoparticles tend to form agglomerated structures, where the degree of agglomeration could be controlled by the use of surfactants. For instance, Tipsavat et al. synthesized magnetite nanoparticles using the solvothermal method, wherein poly(vinyl)pyrrolidone (PVP) was used as the surfactant at different concentrations to study its effect on the performance of the magnetite nanoparticles for electrochemical application [44]. They observed that the particle size of the agglomerates increased with an increase in the PVP concentration, with sizes ranging between 50–90 nm, confirmed by Transmission Electron Microscopy (TEM). Similarly, Klencśar et al. prepared magnetite nanoparticles through the co-precipitation method, with malic acid as the coating agent [45]. Using TEM, they also observed the propensity of the individual magnetite nanoparticles to form agglomerates that are larger in size, diameters of which were 200–500 nm. The above two studies highlighted the exogenous use of surfactants to control the size of magnetite nanoparticles [46]. The utility of surfactants could be prevented if the stoichiometry of the precursors is appropriately optimized, which was performed by Di Iorio et al. [47]. They employed the co-precipitation method for the preparation of magnetite nanoparticles where a synthesis parameter, *p*, was defined, which refers to the ratio of the base concentration to the precursor concentration [48]. Scanning Electron Microscopy (SEM) was used to observe the growth of the nanostructures, wherein it was found that the obtained structures were regular octahedrons possessing crystal facets that were well developed, in the sub-micrometer size for high values of *p*, as shown in Figure 3. 

On the contrary, sub-micrometer grains with a pseudo-spherical morphology and partial agglomeration were observed for low values of *p*. It is to be noted that *p* was varied between 0.13–0.53 in the above study. This observation could be attributed to the relative availability of hydroxyl molecules to precursor molecules, where in cases of low *p* values, the precursor concentration supersedes, resulting in the pronounced oxidation of magnetite to maghemite. 

It is worth nothing that the lattice parameters of the spinel structures of magnetite and maghemite are non-distinguishable when characterized using conventional characterization techniques such as X-Ray Diffraction (XRD), Fourier Transform-Infrared Spectroscopy (FT-IR) [28]. Hence, Mossbauer Spectroscopy is used as one of the available techniques for differentiating the two iron oxide phases by the Zeeman effect, because the Mossbauer spectrum of maghemite is a sextet and asymmetric, where the Fe^3+^ ions in the A and B sites of the spinel have similar Mossbauer parameters [29]. The observed asymmetry is due to very small differences at the A and B sites with respect to the magnetic hyperfine fields and isomer shifts, resulting in line 6 being relatively broader, with reduced intensity, than line 1, as seen in Figure 4B. On the other hand, in the magnetite with an inverse spinel structure, although ferrous and ferric ions are present in equal amounts, the effective valence state remains Fe^2.5+^. This is attributed to fast electron hopping between the ions as their atomic orbitals overlap with each other. In magnetite, the isomer shift is larger because of its greater magnetic hyperfine field in the B sites compared to the A sites, which can be easily recognized in the Mossbauer spectrum, as observed in Figure 4A [30]. In contrast to maghemite, magnetite possesses a sextet Mossbauer spectrum that is not well separated [45]. This could be attributed to the non-stoichiometric distribution of Fe^2+^ and Fe^3+^ ions in the magnetite crystal at any given time, which results in the development of strains in the electronic state of the iron ions present in the octahedral sites. Consequently, the Fe^2.5+^ ions present at the octahedral sites of an ideal magnetite inverse spinel structure are transformed into Fe^3+^ ions.

Electron Magnetic Resonance (EMR), in addition to the Mossbauer Spectroscopy, can be used to differentiate magnetite and maghemite. This is because magnetite nanoparticles exhibit broad ferromagnetic resonance, the EMR spectra of which is asymmetric, as is evident in Figure 5. This asymmetry occurs because the spectra is shallower at the minimum points [45]. This form of EMR spectra is inherent for magnetic crystals that display negative cubic magnetocrystalline anisotropy (MCA), akin to magnetite [50]. The structure and composition of a given crystallite dictates its ability to manifest MCA. Considering magnetite, MCA is caused as a result of its limited spin-orbit coupling and its ferrous ions having a high spin configuration [51]. 

Hence, a decrease is expected to be observed in the MCA for non-stoichiometric magnetite where the number of ferrous ions is less than that of ferric ions. In addition, a particle size dependence of MCA exists where the magnetic moment of small magnetite nanoparticles is subjected to relaxations that are thermally induced. As a result, the volumetric MCA decreases with an increase in the relaxation rates, thereby attaining zero in the superparamagnetic state. This contrasts with large magnetite nanoparticles, where the relaxation phenomenon is highly suppressed, and they thereby exhibit a higher cubic MCA field [52].

The characteristic peaks of magnetite from FT-IR are observed at 580 cm^−1^ and 385 cm^−1^, which are due to the stretching vibration of the Fe-O functional groups and the torsional vibration of Fe in the octahedral site, as observed in Figure 6A [53]. 

X-Ray Photoelectron Spectroscopy (XPS) can also be helpful in distinguishing magnetite from its counterparts. When there is a peak shift to the lower binding energy and broadening due to the presence of both divalent Fe (2*p*_3/2_) and trivalent Fe (2*p*_3/2_), it can be confirmed that it is magnetite rather than maghemite and hematite, which would contain only Fe^3+^ and can be confirmed by a satellite peak at 719.0 eV. This satellite peak is not observed in cases of magnetite because the satellite peaks from Fe^2+^ and Fe^3+^ are lower in intensity, thereby producing broader bands that are flatter and disappear along with background noise, as observed in Figure 6B [53]. 

### 3.2. Hematite

The second most significant iron oxide is hematite (α-Fe_2_O_3_), which is regarded as the most stable iron oxide under ambient conditions because of its thermodynamic stability. It is to be noted that hematite is isostructural with corundum (α-Al_2_O_3_). Hematite consists of Fe^3+^ ions arranged densely in octahedral coordination with the oxygen atoms in hexagonal closed packing with lattice constants, a = 5.0346 Å and c = 13.752 Å, as shown in Figure 2b. The structure can be visualized where Fe^3+^ ion sheets are stacked between two closed-packed layers of oxygen, which are held together by covalent bonds. In addition, hematite’s crystal structure possesses a three-dimensional framework which is linked to 13 neighbors by a face, three edges and six vertices. As Fe is present in a trivalent state in hematite, each of the oxygen atoms is bound to two Fe ions and two oxygen octahedrons are occupied out of the available three. As a result of this arrangement, hematite remains neutral with no deficiency or excess of charge present [4]. Hematite is also shown to exhibit spin canting, a phenomenon where a tilt is produced in the spins of the atoms by a small angle around their axis, instead of being co-parallel. This happens because of two conflicting factors; the isotropic exchange that tends to align the spins exactly anti-parallel and anti-symmetric exchange (due to spin-orbit coupling) that tends to align the spins at exactly 90° to each other [54]. 

The magnetic behavior of hematite nanoparticles is deeply influenced by a handful of parameters, such as the particle size and shape, particle structure, the crystallinity of the particle, its morphology, cation substitution ability and the induction of dipole-dipole and exchange interactions. Considering the shape of the nanoparticles, an interesting study was performed by Tadic et al. in which the magnetic properties of hematite were estimated using three main parameters associated with its shape: namely, its circularity, elongation and convexity [55]. 

Circularity is defined as the degree of difference of a given shape from that of a perfect sphere [56,57,58]. This parameter could take any value between 0 and 1, where 0 is attributed to a line-like structure, while 1 is attributed to a perfect circle. It is mathematically given by: (1)$$Circularity=\frac{4\pi A}{P^{2}}$$
where *A* refers to the area of the shape and *P* refers to the perimeter of the shape.

Elongation, the next shape parameter, could be calculated from the orientation, where the latter is defined as the line that minimizes the integral of the square distances between the points and axes of the shape. Elongation could take up any values between 1 and ∞, where the circle is defined to have the minimum possible elongation of 1 [57,59]. Finally, convexity is measured by the extent of the difference of a given shape from a convex shape and could take up values between 0 and 1, akin to circularity. It is mathematically given by: (2)$$C\;\left(S\right)=\frac{Area\;\left(S\right)}{Area\;\left(CH\left(S\right)\right)}$$
where *S* is a planar shape, *Area* (*S*) refers to the area of the shape and *Area* (*CH*(*S*)) represents the area of the convex hull of S [59,60]. 

The study compared three samples, S1, S2 and S3, which possessed different precursor and surfactant concentrations. Using SEM and TEM images, as shown in Figure 7, it was observed that the samples vary in their morphologies; where S1 had irregularly shaped particles, S2 had plate-like particles and S3 had ellipsoidal shaped 3D nanoparticles. 

Incorporating the shape parameters, S3 was found to have smaller circularity and larger elongation values than those expected of an ellipsoidal structure, while S2 possessed the most compact shape. The magnetic properties of these samples were evaluated using a Vibrating Sample Magnetometer (VSM), in which the shape anisotropy was investigated, and it plays a role in deciphering the magnetic behavior of the hematite nanoparticles. S2 displayed weak ferromagnetism with larger coercivity than S1 due to the presence of shape anisotropy in the former. S3 manifested much larger coercivity and weaker ferromagnetism than S2, approximately 1.5 times that of the bulk hematite and 35 times that of S1. It is to be noted that such a shape anisotropy translates into magnetoelastic anisotropy, where it is caused by defects in large crystals or by internal strains in fine particles. High coercivity is observed in the polycrystalline nanostructures of hematite where the effect of the particle alignment and the exchange of interactions between grains and grain boundaries are enhanced, which explains the elevated coercivity values of the S3 hematite nanoparticles [55]. Similarly, Trpkov et al. synthesized hematite nanoparticles through the hydrothermal strategy, wherein exotic morphologies, such as mushroom-like, cube-like and sphere-like micro-sized superstructures, were obtained using glycine as the surfactant. They observed that when the precursor and glycine were taken in a 1:1 ratio, the resulting hematite nanoparticles exhibited a high coercivity of 5575.6 Oe. On the contrary, when the ratio was increased to 1:12, the resulting nanoparticles exhibited a drastic ~100 times decrease in its coercivity value (42.7 Oe). Such an observation is vindicated by the increase in the resistance of domain rotation [61]. 

From the FT-IR spectrum, hematite can be distinguished from the other iron oxides by calcinating it at a high temperature, around 873–1073 K. The absorption bands of hematite are observed at 540 cm^−1^ and 460 cm^−1^, corresponding to the stretching vibration mode, and bending vibration mode of Fe-O, respectively, as is evident in Figure 3A [62]. Other characterization techniques, such as XPS and Mossbauer spectroscopy, can also be used to identify hematite distinctly: in XPS, a satellite peak is observed at 719.0 eV, indicating the lone presence of Fe^3+^ ions, as observed in Figure 3B; in Mossbauer spectroscopy, spin canting can be observed [53]. X-Ray Fluorescence Spectroscopy (XRF) could also be used to confirm the presence of pure hematite nanoparticles. The characteristic K_α1_, K_α2_ and K_β_ peaks of hematite iron correspond to 6.40384 keV, 6.39084 keV and 7.05798 keV, respectively [63]. 

The Raman spectrum of hematite can provide insights into the vibrational characteristics of the crystal structure of hematite. Typically, the vibrational modes of hematite at the first Brillouin zone are furnished by the equation: (3)$$\Gamma_{ab}=2A_{1g}+2A_{1u}+3A_{2g}+2A_{2u}+5E_{g}+4E_{u}$$
where the acoustic modes $A_{1u}$ and $A_{2u}$ are silent, optically, while the symmetric ones are active by Raman and anti-symmetric by infrared. 

As hematite has a hexagonal crystal structure with an inversion center, the absence of any mode that is both infrared and Raman can be confirmed. Hematite exhibits seven peaks at 229, 500, 249, 295, 302, 414 and 615 cm^−1^; where the first two are attributed to the $A_{1g}$ mode, the remaining are attributed to the $E_{g}$ mode, as is evident in Figure 8A. It is to be noted that, with an increase in the particle size of the iron oxide nanoparticles, the intensity of the Raman scattering increases. This is because, if the number density of the interacting media is high, higher numbers would be the scattering centers in the sample, resulting in an increased scattering intensity [64]. 

X-band Electron Spin Resonance (ESR) stands as another characterization method by which hematite could be discerned from other polymorphs of iron oxide [63]. This is achieved by using the equation that furnishes the Lande’s g-factor: (4)$$g_{k}=\frac{h\nu_{M}}{\mu_{B}H_{K}}$$
where $\mu_{B}$ refers to Bohr’s magnetron, *K* represents the index of the absorption peak and *h* refers to the Planck’s constant. The g-factor for the electron in a single crystal hematite where ferric ions have a *d*^5^ configuration is close to 2.0023 [65]. However, in the hematite nanoparticles prepared by Abd El Aal et al. using the microwave assisted synthesis method, two distinct absorptions were observed with g-factors: g_1_ = 2.022 and g_2_ = 2.424. g_1_ corresponds to absorption by ferric ions and this modest increase in the g-factor could be attributed to the coordination interaction of the oxygen atoms with ferric ion centers at the octahedral sites [66,67]. On the other hand, the positive shift in the g_2_ is due to an electron transfer onto the central hydrogen ion, which acts as an electron acceptor [68,69]. In addition, the absorption widths were different, owing to the differences in the relaxation times of the hydrogen and ferric ions (both spin-spin and spin-lattice), wherein the width of the absorption and relaxation time enjoyed an inverse relationship [63]. Furthermore, it was found that Jahn-Teller distortion could be induced to the octahedral symmetry of hematite if interstitial hydrogen ions were present. Jahn-Teller distortion refers to the induction of a geometric distortion in a nonlinear molecule for reducing the energy and symmetry of the system. This kind of distortion is predominantly observed in octahedral systems and depends strongly on the electronic state of the system. By this phenomenon, either the degeneracy or stability of the molecular system is compromised for the benefit of the other. For instance, if a molecular system is present in a degenerate state, it remains in an unstable state. Consequently, the system undergoes distortion in the form of elongation or compression, where the system’s energy and symmetry are lowered, thereby removing the degeneracy it had possessed earlier. As a result, the spin-lattice relaxation time is increased [63]. 

### 3.3. Maghemite

Maghemite is the next most paramount iron oxide, which is the second most stable polymorph after hematite. Being an allotropic form of magnetite, it shares isomorphous crystal structures with magnetite (edge length, a = 8.346 Å), as shown in Figure 2c. It is also the completely oxidized polymorph of iron oxide, as all the Fe atoms in the crystal are in their Fe^3+^ state. It possesses a spinel crystal structure that maintains a charge neutrality of the unit cell due to the presence of cation vacancies, where the unit cell has 2 and 1/3 vacancies. The distribution of these vacancies also determines the crystal structure in which maghemite would exist. If the vacancies are randomly distributed, which is the general state of this compound, then maghemite has an inverse spinel structure [4]. With the exception of the randomly distributed crystal structure, maghemite can exist in an ordered structure, in which the vacancies are partially ordered. In addition, it can also exist in a tetragonal structure with a threefold doubling, with perfectly ordered vacancies [70]. This polymorph exhibits ferrimagnetism against the anti-ferromagnetism exhibited by hematite. As stated earlier, the differentiation of magnetite and maghemite is difficult unless sensitive characterization techniques are employed. From the Mossbauer spectrum of maghemite, with the use of large applied magnetic fields, the A site and B site hyperfine fields could be discerned with the help of the direction of orientation with respect to the magnetic field, wherein the former aligns parallel, and the latter aligns anti-parallel. Hence, the magnetic splitting of A is observed to increase, while that of B is suppressed, which solves the problem of not being able to distinguish the components of the spectrum due to ferric ions in the A and B sites possessing Mossbauer parameters that are similar [54]. 

From the FT-IR spectrum, maghemite can be ascertained from the other polymorphs of iron oxides. The characteristic peaks of maghemite are obtained at 570 and 410 cm^−1^, as is evident in Figure 6A. The former peak corresponds to the bending vibrational mode of the Fe-O bond at the octahedral and tetrahedral sites and the latter peak to the Fe-O deformation, at the octahedral sites alone, in a disorder maghemite spinel structure [71]. XPS can also be used to identify maghemite, along with hematite, as a clear satellite peak is observed at 719.0 eV for both the iron oxides, highlighting the presence of Fe^3+^ ions in the crystal structure, as shown in Figure 6B [53]. Raman studies can also be performed by heating the iron precursor to the appropriate temperature. This is significant because, at a low temperature, maghemite may not be formed as a result of oxidation or, at a high temperature, hematite contamination may occur. Similar to hematite, the vibrational modes of maghemite at the first Brillouin zone center is given by: (5)$$\Gamma_{ab}=A_{1g}+E_{1g}+T_{1g}+3T_{2g}+2A_{2u}+2E_{u}+4T_{1u}+2T_{2u}$$
where all the modes except $T_{1g}$, $A_{2u}$, $E_{u}$ and $T_{2u}$, will be observable. The symmetrical modes, $A_{1g}$, $E_{g}$ and $3T_{2g},$ are active to Raman, while the $4T_{1u}$ mode is active to infrared. It is to be noted that the frequency of the Raman active phonon modes changes with the synthesis method of maghemite, which influences the vacancies distribution in it. This polymorph has three Raman peaks at 365, 511 and 700 cm^−1^, as seen in Figure 8B [64]. 

### 3.4. β-Fe_2_O_3_

The polymorphs β-Fe_2_O_3_ and ϵ-Fe_2_O_3_ are less prominent and are rarely observed in nature. Although they are occasionally observed, it is worth discussing some points about them. First, β-Fe_2_O_3_ has a cubic body-centered crystal structure as seen in Figure 2d, with an edge length (a) of 9.393 Å. There are two non-identical octahedral cationic sites in the structure, which are named the b and d sites. These sites are occupied by high spin state Fe^3+^ ions (S = 5/2) [4,70]. It did not find any useful applications earlier but recently has been used as a chemical sensor for chloroform [72] and recognized as a potential candidate for the anode material in Lithium-ion batteries [73]. Conventional characterization techniques, such as XRD, XPS, FT-IR and Raman, yield characteristic information about β-Fe_2_O_3_. XRD studies revealed a rhombohedral geometry to be the predominant crystal structure for β-Fe_2_O_3_ [72,74,75]. Considering its XPS spectra, the O1*s* spectrum highlights the lattice oxygen presence by the peak at 535.5 eV [76], while the Fe2*p* spectrum discerns the spin orbit peaks of Fe2*p*_3/2_ and Fe2*p*_1/2_ at 718.6 and 731.8 eV, respectively [77]. In terms of the FT-IR spectra of β-Fe_2_O_3_, a characteristic band was observed at 581 cm^−1^, corresponding to Fe-O-Fe stretching [72]. Intense bands could be observed in the Raman spectrum of β-Fe_2_O_3_ at 289 and 330 cm^−1^, corresponding to Fe-O stretching [72]. 

### 3.5. ϵ-Fe_2_O_3_

On the other hand, ϵ-Fe_2_O_3_ is a recently identified polymorph and possesses an orthorhombic crystal structure, governed by the lattice parameters (a = 5.072 Å, b = 8.736 Å and c = 9.418 Å). Its structure is distinctive as it consists of four oxygen atoms that are packed closely. The structure is beautifully formed, and it consists of edge sharing octahedral triple chains and corner sharing tetrahedral simple chains, each running parallel to the a-axis of the crystal. Its structure consists of six crystallographically non-equivalent anionic and four cationic positions. The cationic positions in the crystal are filled completely with Fe^3+^ ions, with an absence of vacancies in the structure, in contrast to maghemite. The last cationic position is tetrahedrally coordinated, while the first three are octahedrally coordinated [4,70]. It was shown to have the highest coercivity amongst the known metal oxides. This makes it effective for using ϵ-Fe_2_O_3_ as a magnetic recording material in the field of high-density recording media [78]. In addition, it can produce a large coercive field of 2 T at room temperature [70]. 

To discern ϵ-Fe_2_O_3_ from other polymorphs, sophisticated characterization techniques, such as Confocal Raman Microscopy (CRM) [79], resonance Photoelectron Spectroscopy (resPES) [80] and Mossbauer Spectroscopy [81], could be used. The idiosyncratic properties of ϵ-Fe_2_O_3_ identified by each of these techniques will be discussed in detail in the following paragraphs. 

Conventionally, the preparation of ϵ-Fe_2_O_3_ is difficult as it remains unstable in its bulk form. It could be synthesized only in the nano-form and requires the presence of stabilizing agents [82,83]. Hence, synthesis strategies such as the sol-gel method have been predominantly used for its preparation [84,85,86,87]. An exhaustive Raman spectra characterization of this polymorph has been performed by López-Sánchez et al., who had prepared these nanoparticles by the sol-gel route using a silica matrix. They prepared two kinds of samples; namely, powder and thin-film samples, but for discussion’s sake, we shall restrict to powder samples alone. The alkaline hydrolysis methodology was employed to obtain fine powders of ϵ-Fe_2_O_3_ where tetra methyl ammonium hydroxide (TMAH), a strong base and surfactant, was used to promote the ϵ-phase stability [87]. Upon using CMR, individual islands of hematite and ϵ-Fe_2_O_3_ were observed with a homogenous spatial distribution, when micro-Raman mappings were employed [79]. 

As the synthesis of ϵ-Fe_2_O_3_ nanoparticles is challenging, no Raman spectrum has been measured from an isolated phase. As a result, this motivated López-Sánchez et al. to theoretically estimate the number of vibrational modes using the Bilbao crystallographic server, with the help of the Pna2 space group, to which ϵ-Fe_2_O_3_ belongs. They were able to identify 117 vibrational modes that were enabled by the selection rules of Raman scattering, wherein the acoustic modes were not included. Out of the 117 modes, 24 modes were discerned in the range of 100–850 cm^−1^, at room temperature, which corresponds to the first order phonon modes [79]. This group of researchers also performed a comparison of the ϵ-Fe_2_O_3_ expected spectrum with that of hematite. In the case of hematite, seven active phonon modes are present in its Raman spectrum (2 *A*_1*g*_ and 5 *E_g_* modes) namely, *A*_1*g*_ (1) at 228 cm^−1^, *E_g_* (1) at 248 cm^−1^, *E_g_* (2) at 294 cm^−1^, *E_g_* (3) at 301 cm^−1^, *E_g_* (4) at 414 cm^−1^, *A*_1*g*_ (2) at 504 cm^−1^ and E_g_ (5) at 617 cm^−1^ [88]. The paramagnon and two-magnon bands appear at 826 and 1547 cm^−1^ for hematite [89,90]. For ϵ-Fe_2_O_3_ nanoparticles, it was observed that the band positions shifted towards lower wavenumbers, which was plausibly attributed to the smaller compression of the crystal lattice of ϵ-Fe_2_O_3_ and consequent lower phonon confinement, when compared with C. Dejoie et al. [91]. 

Furthermore, a second level of differentiation between hematite and ϵ-Fe_2_O_3_ was proposed by the same group, which involved exposing the samples to increasing powers of the laser light (0.2–30 mW) used in Raman analysis. As the laser light that was used here acted both as an excitation source and probe, in situ structural changes were visualized. It was observed that the vibrational modes of the ϵ-phase shifted towards the lower wavenumbers, as expected, along with an increase in the intensity with a corresponding increase in the laser power, up to 2.8 mW. The modes of the ϵ-phase vanished gradually upon reaching 3.2 mW, whilst that of hematite began to emerge. Between 3.2–3.6 mW laser power, both of the phases co-existed. The hematite vibrational modes began to dominate beyond 3.6 mW, wherein, up to 20 mW, the bands shifted to lower wavenumbers, similarly to the ϵ-phase, but decreased in intensity. Such a shifting behavior could be attributed to the temperature increase induced by the laser irradiation, thereby promoting the anharmonic interactions. Between the 20–30 mW range, the vibrational modes behave the same way, except that the background signal of the spectrum decreases in intensity. This is due to the recrystallization or morphological modification induced by extensive laser irradiation. Hence, appropriate lasers could be used as sources to differentiate between hematite and ϵ-Fe_2_O_3_ nanoparticles [79]. 

The Mossbauer spectrum of ϵ-Fe_2_O_3_ is a superposition of magnetic and paramagnetic components, consisting of quadrupole doublets and Zeeman sextets, where the former corresponds to the iron atoms in the superparamagnetic ϵ-Fe_2_O_3_ nanoparticles [81,92]. The spectrum could be discerned into magnetically ordered portions, represented by three sextets ascribed to Fe1-Fe2, Fe3 and Fe4, which correspond to the iron ions present at the octahedral and tetrahedral sites [93]. The Fe1 and Fe2 sites possess similar hyperfine structural parameters and, hence, remain indistinguishable, while that of Fe4 is well discernable due to a significant variation between the isomer shifts and hyperfine splitting.

Resonant Photoelectron Spectroscopy, a characterization technique that makes use of the synchrotron radiation for the purpose of resonant excitation performed at the absorption edge of a material, is essential for the identification of ϵ-Fe_2_O_3_ nanoparticles for its spin states and satellite structures. The resonant inelastic X-ray scattering prevalent in resPES studies is explained using the Kramers-Heisenberg (KH) situation, which furnishes the expression for photon scattering by an electron in the atom [94]. This situation claims that when resonant excitation occurs, the photo-excited electron (PEE) is transferred to the conduction band of the material, where it holds the ability to interact with all the valence bands and conduction band carriers, within its lifetime. This state of the system is described to be the KH intermediate state whose lifetime equals the time scale of the interaction of PEE with the valence and conduction band carriers. 

Considering the O1*s* profile of this resPES study, a single peak was observed at 531 eV for ϵ-Fe_2_O_3_, while the exact same peak was split for hematite, owing to the differences in the Fe-O bond lengths in the latter. However, in the former polymorph of iron oxide, even though Fe-O bond length differences persist, they are equalized by elevation in the number of covalent bonds between Fe3*d* and (O2*p*) states in the valence band [80]. From the Fe2*p* profile, it can be seen that the main signal in the case of ϵ-Fe_2_O_3_ comes from the high spin configuration of the Fe 3*d*^5^ state, with very low contribution from the low spin states at 711 eV, with a pronounced satellite peak at 719.5 eV. The exact opposite is true for hematitel where the contribution from low spin states is high. In addition, maghemite could also be discerned by the Fe2*p* profile of the resPES study, wherein the satellite emission peak at 719.5 eV is weak or disappears [80]. This observation is corroborated by the concept of intra-atomic 3*d*-4*s* excitation, involving a two-electron correlation effect that is explained by the Eastman-Chiang (EaCh) model [94]. The model claims that the photo-excited electron results in a two-electron process that subsequently removes two electrons from the 3*d* state and fills the lowest state of the conduction band in the 4*s* state of the metal. This transition results in a repulsive Coulombic repulsion between the additional electrons residing in the photo excited state, which is essential to recompense for the spin-spin interactions in the lowest conduction band as the second electron could possibly lower the correlation energy by opting for a paired spin configuration. As a result, strong polarization effects could be seen in the photo-excited state. 

On the contrary, ϵ-Fe_2_O_3_ facilitates weak polarization as it involves only one *d*-*s* excitation, wherein a single electron exists in the conduction band as a result of thermal excitations. This electron is a Wannier exciton, which is highly delocalized in the crystal lattice, in contrast to Frenkel excitons, which are localized. Wannier excitons establish the screening of the electric field to achieve a reduction in the Coulomb interactions [80]. In addition, it is known that a single electron correlation satellite is more likely to be excited than two electron correlation satellites, which explains the intensity of satellite emission being greatest for ϵ-Fe_2_O_3_, followed by hematite and then maghemite. Maghemite possesses the least intensity, owing to the homogenous distribution of bond lengths resulting in the quenching of intra-atomic *d*-*s* coupling [80]. 

### 3.6. ζ-Fe_2_O_3_

In addition to the above polymorphs, a new polymorph of iron oxide has been identified, which is named ζ-Fe_2_O_3_. β-Fe_2_O_3_ is transformed into ζ-Fe_2_O_3_ at a high-pressure (>30 GPa). Surprisingly, ζ-Fe_2_O_3_ exhibits room temperature stability even after the relaxation of the pressure. It is characterized as having a monoclinic crystal structure, as seen in Figure 2e, and exhibits anti-ferromagnetism. Its stability at room temperature is attributed to the high surface energy it possesses from smaller β-Fe_2_O_3_ nanoparticles. The structural features of ζ-Fe_2_O_3_ are quite different from that of β-Fe_2_O_3_ because of pressure-induced effects, although β-Fe_2_O_3_ is the precursor. The octahedral Fe b-site of β-Fe_2_O_3_ splits into two non-equivalent Fe sites in ζ-Fe_2_O_3_ and the octahedral Fe *d*-site splits into four non-identical Fe sites. XRD can be used to distinguish ζ-Fe_2_O_3_ from β- Fe_2_O_3_ as the symmetry of the crystal is lowered from cubic to monoclinic [5]. 

### 3.7. Goethite

Iron oxyhydroxides are the next class of iron oxide polymorphs. Goethite is the most commonly found oxyhydroxide of iron, possessing an orthorhombic structure characterized by the lattice parameters, a = 9.95 Å, b = 3.01 Å and c = 4.62 Å. It possesses a 3-D structure, formed as a result of tunnels formed from FeO_3_(OH)_3_ octahedra and spreading in the (0 1 0) direction, where the hydrogen atoms are positioned, as can be seen in Figure 2f. This crystal structure is peculiar, where a single octahedron is coordinated with eight octahedral neighbors, by four edges and three vertices, with the oxygen atoms present in the tetrahedral surrounding environment [4]. Goethite nanoparticles are generally rod-shaped and lack crystalline order due to the presence of a large number of grains in the particles and grain boundaries of smaller angles. The Mossbauer spectrum of goethite at low temperatures remains split magnetically with a magnetic hyperfine field of 49.5 T and an isomer shift of 0.49 mm s^−1^. However, as the temperature is raised (>220 K), asymmetric line broadening is observed because the magnetic properties are affected by the fluctuations in the magnetic field that perturb the interacting grains of goethite [95,96]. 

### 3.8. Ferrihydrite

The next polymorph of iron oxyhydroxides is ferrihydrite. It has a poor crystalline structure, as evident from Figure 2g, and its particles have a smaller size of around 2–6 nm. Due to its smaller size, it finds application as a catalyst and adsorbent of toxic ions and as the core for ferritin protein, which sequesters and stores iron in plants and animals [4]. The XRD patterns of ferrihydrite consist of few, broad reflections that corresponds to the 6-lines ferrihydrite and 2-lines ferrihydrite. The 6-lines ferrihydrite possesses more order and its XRD spectrum consists of six well defined peaks. On the other hand, the 2-lines ferrihydrite is randomly ordered or poorly crystalline, possessing only two broad peaks in its XRD spectrum. As the different synthesis procedures of ferrihydrite creates variable crystalline nature and water content, there are different chemical formulae for ferrihydrite, but the most reported one is 5Fe_2_O_3_.9H_2_O. The precise and exact atomic structure of ferrihydrite remains a mystery and has been frequently debated. However, the above two kinds of ferrihydrites differ in their magnetic properties, where the 6-lines type is anti-ferromagnetic but, at low temperatures, exhibit super-paramagnetism. It has also been found that the magnetic properties of the 6-lines type depend on the particle size. The 2-lines type, on the contrary, due to the presence of uncompensated spin moments at the surface of these nanoparticles, exhibit ferromagnetism [97]. 

### 3.9. Wüstite

The next most vital iron oxyhydroxides polymorph is the Wüstite. It is characterized by a cubic unit cell crystal with an edge length of a = 4.239 Å, as observed in Figure 2h, which remains stable at high temperatures (>843 K) and low pressures. Wüstite has a rock-salt type crystal structure. The presence of large oxygen anions in the crystal forms a close packed face centered cubic sub-lattice such as the spinel type magnetite and maghemite with the small sized Fe^2+^ ions populating the interstitial sites. It is observed that almost all of the Fe ions are bound to oxygen in an octahedral coordination. The identity feature of this polymorph is that the (1 1 1) planes of iron and oxygen form an ideal 2-D hexagonal lattice, the inter-atomic distance of which is 0.304 nm, which corresponds to the hexagonal unit cell of FeO. Wüstite can be oxidized to form magnetite or maghemite, through long time storage in open air, where the spinel-type structures become dominant over the rock-salt structures with an increase in storage time. This modification is a result of the aggregation of a basic cluster containing four divalent Fe vacancies at the octahedral sites surrounding each trivalent Fe ion located at the tetrahedral interstitials [4]. 

### 3.10. Akaganeite

Akaganeite is one of the iron oxyhydroxide phases with the largest tunnel-type structure amongst the iron oxide phases. It is a natural product of the iron corrosion process in environments containing chloride. It has a tetragonal crystal structure containing double chains of edge-shared octahedra, which shares its corners with adjacent chains to form the tunnel structure, as can be seen in Figure 2i. As it is formed in chloride (Cl) containing environments, chloride is present as an impurity in the structure. Here, FeH occupies the octahedral sites along with the Cl and water located in the tunnels. The presence of Cl in the octahedral site is justified by the requirement of halide anions in the structure to balance the excess proportion of oxides in the iron octahedra and the acidic environment in which the structure is formed. As a result, akaganeite stands as a suitable material for catalysis and ion exchange [4]. Akaganeite has two characteristic peaks in FT-IR, at 845 and 685 cm^−1^ which indicates the hydrogen bonding between H and Cl in chloride-containing akaganeite. In addition, a split band is observed at 3440 and 3340 cm^−1^ in O-H stretching region, which again indicates the presence of O-H stretching bands in akaganeite. From their XRD spectrum, it can be found that they generally tend to be present as rod-like crystals because the intensity of the (2 1 1) peak is greater than that of (3 1 0) [98,99]. 

### 3.11. Lepidocrocite

The final polymorph of iron oxyhydroxides that will be discussed is Lepidocrocite. It is predominantly found in hydromorphic soils where the seasonal alteration of reducing and oxidizing conditions is present. It acts as a precursor for magnetite and maghemite. Possessing an orthorhombic structure, it is assembled by iron octahedra and hydroxyl group double layers that occupy their external surfaces, as seen in Figure 2j. The assembly is formed as a result of hydrogen bonding between the hydrogen and oxygen atoms present in the octahedra and hydroxyl groups. It is hypothesized that the hydrogen atoms would be present at the inversion centers, and it is equidistant from two oxygen atoms of the adjacent layers. This sort of arrangement provides continuous O-H-O-H-O chains that possess hydrogen bond symmetry. These nanoparticles usually have a lath-like or tabular morphology [4]. It finds its purpose as an appropriate bio remediating material because it is used in the remediation of water contaminated with carbon tetrachloride [100]. 

## 4. Properties of Iron Oxide Nanoparticles

To engineer any material for a specific application, the desired properties must be imparted to the material and are expected out of it. Hence, properties of the material remain as the next indispensable factor, after structure, in designing a material for a specific function. Towards this end, the different properties exhibited by the ubiquitous iron oxide nanoparticles will be discussed in this section. Initially, the optical properties will be discussed, followed by the magnetic and rheological properties of the iron oxide nanoparticles. 

### 4.1. Optical Properties 

The optical properties are essential because they are used to design electro-chromic devices, as well as for the photo-electrochemical generation of hydrogen and solar radiation filters [101]. He et al. [102] suggested that the intensity of absorbance by iron oxide nanoparticles grossly varies with the particle size. As the particle size increases, the tendency of the nanoparticles to aggregate increases, thus increasing the absorbance band. It has also been observed that heating the nanoparticle samples influences the band gap, as the crystal structure changes with heating. In addition, doping the iron oxide nanoparticles with other elements can induce a lattice strain, influencing the band gap of the nanoparticles. The work conducted by Nair et al., and Malik et al. reveals that, after doping, a red shift is observed in the absorbance, indicating a pressure-induced effect. When the particle size is small (achieved by doping), the surface pressure increases, which increases the lattice strain, resulting in a decrease in the band gap. On the other hand, in cases of undoped iron oxide nanoparticles, the band gap is increased because of the weak exciton confinement [103,104]. 

The particle size, in cases of iron oxide nanoparticles, is observed to decrease with an increase in the annealing temperature, up to 673 K, and then increases after 715 K. This is because the hematite and maghemite phases are present below 673 K and maghemite disappears above 715 K. As a result, the band gap decreases up to 673 K because the lattice strain is higher due to the presence of the two phases. After 715 K, due to the presence of hematite alone, the lattice strain becomes minimal, making the band gap minimum too [105]. Magnetite and Wüstite exhibit thermally induced electronic transitions attributed to the inter-valence charge transfer, because of which they show absorption in the visible and near-IR region. The magnetite to maghemite nanoparticles’ conversion due to oxidation can be tracked by a loss in the optical absorption near-IR region [106,107]. Hematite has a direct optical band gap between 2.0–2.7 eV (determined from Tauc plots as shown in Figure 9a,b) and an indirect band gap of 1.9 eV [108]. For amorphous β-Fe2O3 nanoparticles, direct band gaps of 1.73 and 1.97 eV have been determined. Maghemite nanoparticles have a band gap of 2.47 eV against the band gap of bulk maghemite of 2.0 eV, which is due to quantum confinement effects [101].

### 4.2. Magnetic Properties

Integral parameters, such as magnetic susceptibility, permeability, magnetic moment, and magnetization, are useful to characterize the magnetic properties of solids. With respect to iron oxides, the primary interaction between the iron ions present in the adjoining sites is the electrostatic exchange interaction. This interaction results in iron oxides enclosed by oxygen atoms or hydroxyl ions so that the exchange can take place through the ligand present in between. At the molecular level, electrons in the eg orbitals of Fe^3+^ ions, which are unpaired magnetically, interact with those electrons in the p-orbitals of the oxygen atoms, only upon the situation in which the cation and ligand are in critically closer distance to allow the coupling of their electrons. This gives rise to a chain coupling effect that results in percolation through the crystal. It is to be noted that the exchange constants for the above process is dependent on bond parameters, such as the bond length and angle, between the cation and the ligand. The interactions are strong when the bond angles are between 120–180° and become weak when the angle is 90°. 

In magnetite, as both Fe^2+^ and Fe^3+^ ions are present, electrons delocalize between the adjacent sites occupied by the iron ions. Sophisticated characterization techniques, such as Mossbauer Spectroscopy, Neutron Powder Diffraction, Vibrating Sample Magnetometry, could be employed to evaluate the magnetic performance of the synthesized iron oxide nanoparticles. In order to measure the characteristic parameters associated with the magnetic properties of any material, a Superconducting Quantum Interference Device (SQUID) magnetometer could be used, which is based on the Josephson Effect. Usually, magnetic moment, magnetic anisotropy constants, saturation magnetization, permeability, hysteresis loops and magnetic hyperfine field are the unique parameters that are considered for the magnetic characterization of any given material. The inner s electrons of the iron atom are polarized by the 3d electrons, resulting in a magnetic hyperfine field [4,6,33]. 

The particle size of the iron oxide nanoparticles influences the magnetic properties to a great extent. Patsula et al. [109] observed that there is a transition from super-paramagnetism to ferrimagnetism when the particle size is increased (three samples whose size ranged between 10–24 nm). Super-paramagnetism was attributed to the samples with a lower particle size because of the observed low values of coercivity (H_c_) and relative remanence (M_r_/M_s_) in comparison to the relatively higher values of H_c_ and M_r_/M_s_ in samples with the largest particle size showing ferrimagnetism. Hence, it can be correlated that coercivity and relative remanence are functions of particle size because the particle agglomeration and Brownian relaxation effects influence them. The hysteresis loops used in this study involved liquid dispersions, wherein the properties of these dispersions influence heating power of the fluids. In addition, the specific absorption rate for the ferrimagnetic sample is higher compared to the superparamagnetic sample, indicating higher reversal losses in the former than the latter [109]. 

Ultra-small iron oxide nanoparticles exhibit different kinds of magnetic properties. When the zero-field-cooled (ZFC) and field-cooled (FC) magnetic susceptibilities for maghemite were measured by Milivojevic et al. [110] as a function of temperature, the magnetizations bifurcated at a certain temperature, which is a characteristic of spin-glass systems or interacting and non-interacting superparamagnetic nanoparticle systems. Fluid ultra-small maghemite nanoparticles and powder nanoparticles of maghemite exhibit different magnetization behavior. In powder nanoparticles, the FC magnetization decreases and flattens with the decrease in temperature below T_B_ (blocking temperature below which magnetic moments are blocked). This is a property of interacting superparamagnetic particles with strong interaction and of spin and super-spin glass. On the other hand, with fluid nanoparticles, the FC magnetization increases with the decrease in temperature, which is a property of super paramagnets alone. 

In hematite nanoparticles, the susceptibility is enhanced by five times compared to its bulk counterpart. The increased susceptibility is primarily attributed to the uncompensated surface spins as well as, to a certain extent, the lattice defects. The following is the explanation for the enhancement of magnetic susceptibility. Hematite exhibits an antiferromagnetic order when the temperature is maintained below its Neel temperature (955 K) and becomes paramagnetic above this temperature, as it loses its magnetic ordering. The magnetic sub-lattices of hematite orient along the rhombohedral [1 1 1] axis and are anti-parallel below the Morin transition temperature (263 K). The magnetic moments, on the other hand, gives rise to small net magnetization as a result of their spin canting. With a decrease in particle size, the Morin temperature decreases correspondingly due to lattice expansion in the small particles [62,111]. 

β-Fe_2_O_3_ nanoparticles remain as the only Fe^3+^ oxide polymorph that remain paramagnetic at room temperature as its Neel temperature lies between 100 K and 119 K. It is to be noted that β-Fe_2_O_3_ orders itself anti-ferromagnetically below its Neel temperature [70]. 

The magnetic properties of ε-Fe_2_O_3_ nanoparticles are not completely understood because it exhibits similar properties to both hematite and maghemite. Two magnetic transitions have been reported for this polymorph, namely at 495 K and 110 K, wherein the former corresponds to its Curie temperature. It was found that ε-Fe_2_O_3_ transitions from its usual paramagnetic behavior to a more magnetically ordered state at its Curie temperature. On the other hand, it transforms from this state to another magnetic behavior, which seems to be distinct from what is observed at the room temperature behavior of collinear ferrimagnetism. The magnetic state of ε-Fe_2_O_3_ below 110 K is debated; it has been theorized that it might exhibit metamagnetism. ε-Fe_2_O_3_ exhibits a high room temperature coercivity, which is believed to be due to its disordered structure. In addition to high coercivity, it possesses high magnetocrystalline anisotropy due to its uniform nanoparticle size and non-zero Fe^3+^ magnetic moment (orbital component), justifying the presence of comparable spin-orbit coupling. The distortions at the coordinated polyhedra of Fe_1_ and Fe_2_ (crystallographic nonequivalent cationic sites 1 and 2) sites lead to a couple of phenomena: firstly, the mixing of Fe (3*d*)–O (2*p*); secondly, the charge transfer of O (2*p*)–Fe (3*d*). This phenomenon, in addition to the crystal field effects, enhances the electronic degeneracy of the *d* orbitals, resulting in an electronic state with non-zero angular momentum. However, the high coercivity of ε- Fe_2_O_3_ was hit significantly at 110 K, which can be attributed to a decrease in the spin-orbit coupling in relation to the transformations in the structure and magnetism of ε-Fe_2_O_3_ that accompanies this transition [70,78]. 

ζ-Fe_2_O_3_ nanoparticles have shown a pronounced peak at 69 K, which is the Neel temperature, when susceptibility is plotted against temperature. When moved away from this temperature, the susceptibility decreased, exhibiting a transition to the antiferromagnetic state. The transition at 69 K was found to be a second-order thermodynamic transition owing to the polymorph becoming antiferromagnetic from an earlier paramagnetic state. It is interesting to note that the decrease in the strength of the super exchanging interaction in ζ-Fe_2_O_3_ can be ascribed to its greater lattice volume than β-Fe_2_O_3_ [5]. 

With respect to goethite ultra-small nanoparticles, the fluctuations in the sub-lattice magnetization directions within grains that strongly interact in the particles seems to control the magnetic relaxation, which is why the Mossbauer spectrum of goethite contain sextets with asymmetric broad lines over a wide range of temperatures. In addition, goethite being poorly crystalline with dislocations and grain boundaries of smaller angles, the nanoparticles of goethite can be considered to have interacting grains, due to exchange coupling, which confirms that the magnetic relaxation is controlled by superparamagnetic relaxation [112]. 

Akaganeite nanoparticles exhibit a unique magnetic property of magnetic birefringence when it is coupled with a polysaccharide aqueous suspension. Although akaganeite, in its bulk form, is antiferromagnetic below room temperature, its particle size brings in anomalies. This is because very small antiferromagnetic particles show a net magnetic moment arising from incomplete surface spin compensation when the volume spins are outnumbered by surface spins (surface-to-volume effect) [98]. 

Finally, when the magnetic properties of ferrihydrite nanoparticles are considered, higher magnetization values are obtained for 2-line ferrihydrite than for 6-line ferrihydrite because the 2-line kind has a small nanoparticle mean size. In addition, the 2-line type is more anisotropic than the 6-line kind because of its higher effective anisotropy constant. This is because ferromagnetism originates from the surface uncompensated spin moments, which are located at the surface of the nanoparticles. 

Hence, the one with a higher surface-to-volume ratio, in this case 2-line ferrihydrite, would have higher magnetization than its counterpart, 6-line ferrihydrite [97,98].

### 4.3. Rheological Properties

Magneto-rheological fluids are obtained when non-colloidal suspension is induced with particles that can be magnetized and suspended in a carrier fluid whose rheological properties can be controlled by an external magnetic field. Depending upon the strength of the applied magnetic field, these fluids are capable of transitioning from Newtonian fluids to plastic rheological behavior, in other words non-Newtonian fluids. This effect could be explained as a result of the alignment of the suspended particles along the direction of the magnetic field to form chains. This brings about changes in rheological properties, such as viscosity and yield stress, giving rise to the magneto-rheological (MR) effect. Hence, the extent of the MR effect depends on the suspended particle concentration and the saturation magnetization of the applied field. Carbonyl iron, an example of a MR fluid, has a characteristic high magnetic permeability with soft magnetic characteristics, in addition to its high magnetization value and low remnant magnetization. Although these MR fluids seem provide overwhelming properties, they are sensitive to particle aggregation and settling, which limits their applications for general purposes [35,113,114]. 

Magnetite can also be used as the suspended material in the MR fluid. As similar magnetic properties are observed for both maghemite and magnetite, even with magnetite leading maghemite with its magnetization value, maghemite is usually preferred owing to its chemical stability, which is all that is required for MR fluids. This is because magnetite nanoparticles oxidize to ferric oxyhydroxides due to the presence of Fe^2+^ in magnetite leading to the loss of magnetic properties present in the material. In contrast, maghemite consists of only Fe^3+^ ions and cationic vacancies, providing charge neutrality to the unit cells, which maximizes the homogeneity of the distribution in the suspension. As a result, MR fluids with maghemite as the suspended particles tend to tolerate higher magnetic field strengths prior to the saturation of yield stress [113]. Furthermore, this leads to an increase in the maximum dynamic yield stress with narrow magnetization loops and very low coercivity. 

Concentrating on the rheological parameters of MR fluids with maghemite, the sedimentation rate is decreased due to the increased frictional interactions between suspended particles and the carrier fluid. The frictional interactions are higher because the nanoparticles have a greater surface-to-volume ratio. The rheological flow of these fluids is characterized as a function of the magnetic field [115]. In the absence of the field, they exist as non-Newtonian fluids because of the inter-particle interaction and remnant magnetization present in the suspended magnetic particles. However, in the presence of the applied field, shear-thinning behavior is exhibited where the shear stress increases with the increase in the magnetic field. This can be explained by the induced effect of the construction of columns within the fluids, which are formed as a result of strong dipole-dipole interactions. These columns are sensitive to shear stress, which are deconstructed when the shear rate is increased. Lastly, the increase in the dynamic yield stress is attributed to nanoparticles filling the cavities of microparticles; the presence of different sized particles increases the packing fraction and the formation of longer chains when the nanoparticles are present [116]. 

Ferro fluids, on the other hand, are the other extreme of MR fluids. These fluids are obtained as a result of the colloidal suspension of magnetic iron oxide nanoparticles, which can exhibit a magneto-viscous effect due to the interaction between the magnetic moment and mechanical torque of the particles. Similar to MR fluids, the iron oxide nanoparticles that are used extensively are magnetite and maghemite, with oil, water, or kerosene as the dispersing media [34]. Various long and short-range forces can give rise to the sedimentation and agglomeration of the nanoparticles in ferro fluids, and the stability is determined by the steric effect, or the charges carried by the particles. Surfactants are used in order to do away with the steric effect [35]. A good quality ferrofluid is characterized by nanoparticles 4–20 nm in size, with a 2 nm layer of surfactant. The rheological properties of ferrofluids are determined by the ferrofluid flow, in which the magnetic moment of the suspended particles align with the vortices of the flow, initially, and re-orient themselves when a magnetic field is applied [117]. Ferrofluids exhibit non-Newtonian behavior, which can be confirmed from the non-linear decay of viscosity with an increasing shear rate. It is to be noted that the viscosity of the ferrofluids decreases with the increasing shear rate at a constant magnetic field and the viscosity increases with the applied magnetic field at a fixed shear rate [118].

## 5. Influence on Oxidation States and Phase Transformations

Although iron oxides have been shown to exist in sixteen different phases, the crystalline phases are the ones that find their applications. Magnetite and Maghemite have a spinel structure, in which Fe exist in different oxidation states; that is, in both tetrahedral and octahedral geometries [4]. It is understood that the synthesis method has an influence on the distribution of the oxidation state of Fe in the final product. In other words, as oxidation is dependent on the diffusion of oxygen, the degree of oxidation is greatly influenced by the nanoparticle size [6]. 

It is known that when fine magnetite nanoparticles are left at room temperature in air, they oxidize to maghemite. This is because the oxidation leads to a cation-deficient spinel in which the ferrite is modified as [Fe^3+^]_A_[Fe^2+^_1−3 ×_ Fe^2+^_1+2 ×_ X_x_]_B_O_4_, where X refers to the vacancies. Hence, it can be concluded that the quantity of maghemite composition increases in the solution as the nanoparticles size decreases. This is because of the experimental observation in the FT- IR spectrum, where characteristic absorption bands of maghemite are seen around 570 cm^−1^ along with small shoulders in the range 600–750 cm^−1^ [119]. Hematite, on the other hand, can also be transformed to possess increased oxygen vacancies to enhance the stability and rate performance of the anode material used in Lithium-ion batteries. The introduction of oxygen vacancies into hematite modulates the inherent electrochemical properties of the metal oxides and facilitates the phase transition. The presence of increased oxygen vacancies influences the electronic structure of hematite; decreasing the energy necessities electron or ion diffusion and lowers the resistance, leading to improved electrochemical performances of the anode material [20]. 

Thermally induced phase transitions of less thermodynamically stable oxides of iron (β, γ, ε) have been studied extensively. These transitions are highly dependent on the inherent properties of the phase in which the precursor is present, such as in the polymorph structure, particle size, morphology, and particle aggregation. No matter what the precursor or the synthesis method in which the nanoparticles are prepared, these transformations predominantly lead to hematite as the final product, as most of the synthesis methods involve heating, and the stability of hematite is wonderful at higher temperatures. Whilst β-Fe_2_O_3_ and ε-Fe_2_O_3_ can transform directly to hematite, there are exceptions to this. For instance, the powdered nature of β-Fe_2_O_3_ transformed to hematite at 863 K in the due course mechanism of Fe_2_(SO4)_3_ thermal decomposition in air [70]. Interestingly, β-Fe_2_O_3_ dispersed in a silica matrix exhibited higher thermal stability when the transformation temperature was increased to 1473 K [120]. On the other hand, when hollow β-Fe_2_O_3_ nanoparticles are synthesized, they are transformed into maghemite, in contrast to powdered β-Fe_2_O_3_, indicating that the morphology of the precursor is essential for the transformation and allows for unexpected intermediate polymorphs of iron to evolve [5]. 

Hematite, being able to be stable at higher temperatures, is the final product of thermally induced phase transformation or decomposition of most of the Fe^2+^ and Fe^3+^ containing compounds and the ultimate product of thermally induced structural transformations of other Fe_2_O_3_ polymorphs. This makes hematite the most easily synthesizable compound compared to its counterparts. β-Fe_2_O_3_ is the next kind of iron oxide that is available only as nanostructures and not stable in its bulk form. As the structure of β-Fe_2_O_3_ is thermodynamically unstable, it transforms into either hematite or magnetite on heating and even to ζ-Fe_2_O_3_ as a result of high-pressure transformation. Maghemite is converted to hematite either directly or indirectly when the temperature of the reaction condition crosses the threshold value, the value of which depends on various physiochemical parameters, such as the pH, ionic strength of the solution, solvent used, etc. ε-Fe_2_O_3_ can exist as an intermediate during the above conversion [70]. 

Phase transitions induced by thermal or mechanical means are deeply influenced by particle size and doping. At elevated temperatures, the irreversible transformation process of maghemite to hematite depends primarily on the morphology and size of the particles, the material form (powder/thin film/nanocomposite/core-shell structure/surface coated) and the environment under which the transition occurs [70]. It is observed that when the reaction time of the synthesis method by thermal decomposition (at 573 K) is increased, maghemite to hematite transformation was noted. However, at 658 K, nanocrystalline maghemite directly transformed to microcrystalline hematite. It is hypothesized that only when the particle size of maghemite nanoparticles cross the critical size (10–25 nm), the phase transformation occurs [49,70]. Doping is the second most influencing factor towards phase transitions of iron oxide polymorphs. When the maghemite nanoparticles are doped with Zn^2+^ ions, the phase transition temperature is increased by 373 K [121]. In addition, Y_2_O_3_-doped maghemite nanoparticles are found to be thermally stable up to the temperature of 998 K, thus qualifying it to be used as a gas sensor with long-term stability [122]. Apart from particle size and doping, the incorporation of the iron oxide nanoparticles into matrices has been found to affect the phase transition of these particles. Maghemite was found to be thermally stable in a SiO_2_ matrix by some of the authors, who have reported that only above 1173 K, it was transformed to hematite with ε-Fe_2_O_3_ as the intermediate. Here, the porous silica matrix served as an anti-sintering agent that can prolong the life of the intermediate ε-Fe_2_O_3_, resulting in the prevention of the transition to hematite at higher temperatures. Due to this space restriction, the size of the ε-Fe_2_O_3_ nanoparticles is restricted to around 200 nm. On the contrary, in cases of normal powdered samples of maghemite, the decrease in free energy of the nanocomposite system is a result of the reduced surface area of the particles, due to agglomeration and sintering [34,36,123]. 

To conclude the discussion on phase transitions induced by thermal means, it is some points concerning the effect of the precursor towards the transitions are worth mentioning. Generally, the determination of the Fe_2_O_3_ polymorph being formed from a given precursor and its subsequent transformation to other polymorphs depends on two main factors. Firstly, the free energy (G) per unit volume (V); secondly, the energy barrier that the iron oxide phases must overcome to undergo the required transformations. From these factors, the G/V ratio depends on the chemical potential (η) and surface energy (σ). The function is G/V = η/(v + (6σ/d)), where v is the molar volume of the precursor and d is the size of the nanoparticle [78,124]. 

In addition to heat treatment, mechanically induced phase transformations could be conducted through milling processes, laser irradiation and solid-state mixing by applying high pressure. Microcrystalline hematite with size 10 μm undergoes phase transformation to nanocrystalline maghemite of size 12 nm, where hard mechanical grinding using high-energy planetary ball mill was employed [36]. Below the size of 40 nm (critical size for dry hematite nanoparticles by the process employed), the tetrahedral defects of the Fe atoms at the nanoparticle surface promote the formation of maghemite, gradually. This is because of the shearing movement of the oxygen planes in hematite during ball milling, which creates a thermodynamically unstable structural transformation to form the vacant crystal structure of maghemite [70,112]. Recently, as mentioned in the above sections, ζ-Fe_2_O_3_ nanoparticles were successfully synthesized through the high-pressure mechanically induced transformation of β-Fe_2_O_3_ with room temperature stability [5].

A complete summary of the different phases of iron oxide nanoparticles with their chemical formulae, possible stable oxidation states, type of crystal structures, magnetic properties, and the influence of their oxidation states on phase transformations are provided in Table 1.

## 6. Biological Applications of Iron Oxide Nanoparticles

Every advancement in science and technology is intended to make our lives easier and better. Biology, being a part of science, demands major development as it deals with the health of human beings. Nanoparticles play a major role in fulfilling this need for development. Its widespread properties result in vast biological applications. In the subsequent sections, out of the 14 reported polymorphs of iron oxide, magnetite, hematite and maghemite have been utilized as those polymorphs capable of eliciting an intended biological function with the desired biocompatibility. With regards to their physical properties, the tuning of their magnetic properties has been easily achieved in comparison to the other polymorphs, thereby finding their golden utility for biological applications.

### 6.1. Iron Oxide Nanoparticles as Contrast Agents

Iron oxide nanoparticles (Fe_2_O_3_/Fe_3_O_4_) exhibit a superparamagnetic property due to the formation of domains in the micro or nano dimensions that have atoms with unpaired electrons with the same spin. This property creates a magnetic field gradient around the iron oxide nanoparticles, which results in dephasing the neighboring protons, thereby producing a significant T2/T2* relaxation [126]. Iron oxide nanoparticles are used in efficient tissue imaging by magnetic resonance imaging (MRI), in the form of contrasting agents to differentiate normal and diseased tissues. They are generally coated with polysaccharides to maintain bioavailability and prevent immune responses.

Tanimoto et al. prepared superparamagnetic iron oxide nanoparticles (SPIOs) made of crystals of magnetite and maghemite for imaging hepatocellular cancerous cells. The SPIOs were made biocompatible by a coating of dextran or carboxy-dextran. Imaging techniques such as MRI (Magnetic Resonance Imaging) were used and the results were compared using conventional methods, such as non-enhanced MRI [127], dynamic CT [128], dynamic MRI [129,130] with SPIO enhanced MRI. The SPIO-enhanced MRI was found to have better accuracy and mean sensitivity than non-enhanced MRI and dynamic MRI. However, in comparison with the dynamic MRI/hepatobiliary agents-enhanced MRI, SPIO-enhanced MRI slightly lacked accuracy and mean sensitivity [126]. To overcome this, SPIO-enhanced MRI and Gd-based dynamic MRI were used alongside each other to obtain increased accuracy and mean sensitivity than their individual values [131].

Varying sizes of SPIOs can be used for imaging specific tissue types. Iron oxide nanoparticles (IONPs) with sizes less than 30 nm, coated with proteins such as lactoferrin and transferrin were found to have strong affinity for human fibroblast surfaces and showed evasion of immune reactions. IONPs coated with different proteins interact differently as its binding with proteins changes its magnetic moment values. Such a change affects the T2/T2* relaxation time, resulting in varied contrasting abilities of different tissues.

USPIO (Ultra-small Superparamagnetic Iron oxide nanoparticles) were used in cardiovascular MRI (CMR) for the study of myocardial infarct pathology. Multiple scans were taken, at time intervals of 24 h and 48 h post administration of the USPIO and were compared with the ones taken before administration to analyze the efficiency of USPIO as a contrast agent. It was observed that the USPIO-based contrast agents provided better detailed imaging of the myocardial infarct pathology and was superior to the gadolinium-based contrasting agents as the USPIOs detect by infiltrating the macrophages while gadolinium-based detection works with necrosis/fibrosis detection. In addition, these had a negligible effect on the immune system with no renal elimination, thereby making it a safe contrasting agent [132].

In another study, SPIONs were employed as a guidance and MRI-based tracking tool for effective stem cell therapy approaches. The study was performed with different stem cell types, such as bone marrow derived mesenchymal stromal cells (BMSC), neural stem cells (NSC) and hematopoietic stem cells (HSC) and compared with commercially available dextran-coated SPIONs and FeraTrack Direct (FTD) for their respective labeling efficiency. FTD, in comparison to commercially available contrasting agents, are made for easy uptake by cells without the need for any transfection agent [133]. The BMSCs and NSCs were labelled with FTD by overnight incubation and HSC was labelled by incubating it for 24–48 h for intake of FTD. TEM was used to confirm the uptake and encapsulation by endosomes [134] of the stem cells. These cells were tested on tumor-induced nude mice for in vivo analysis. The mice were subjected to three T MRI for a period of 10 days, with the administration of the labelled stem cells each day. The MRI and histological analysis revealed the presence of stem cells of approximately 10 µm-thick sections of tumor. These tumor samples were subjected to nanotoxicity studies, which showed that there were minimal toxic effects on the stem cell functions from the iron oxide nanoparticles. This can be rationalized by the selection of labelled stem cells by magnetically activated cell sorting (MACS), which isolated those stem cells that had encapsulated FTD into the endosomes of the cells. Thus, these cells were found to be unaffected by the SPIONs, making them the right candidate for stem cell-based therapy [133].

In another study, the effect of SPIONs coated on a PLGA particle through the Pickering emulsification method [135] and encapsulated in another by varying the size range of the PLGA particle, and their contrasting ability was observed. Initially, the authors studied the difference in effect of SPIONs coated on PLGA microparticles and encapsulated by one on the MR relaxivity that was studied using a 14 T magnetic resonance (MR) scanner. The results from this test showed that the SPIONs immobilized on the surface of the PLGA microparticles showed three times higher relaxivity than the encapsulated ones.

Next, they varied the size of the PLGA particle from 2 µm to about 260 nm. This size reduction increased the overall surface area of the nanoparticles, thereby providing more surface area for the immobilization of the formed SPIONs onto the PLGA particle surface. The effect of the size reduction was tested again with the 14 T MR scanner. This experiment revealed that the SPIONs immobilized on the surface of PLGA NPs showed an 18-fold increase in the relaxivity in comparison to those SPIONs encapsulated in PLGA NPs. This system was also tested against free SPIONs and found to have around 1.5 times higher relaxivity. Similar results were derived from a 3 T MR scanner. These results were rationalized based on the concepts of “static dephasing regime” and SPION interaction with water. By nature, free SPIONs tend to form clusters that result in an increased particle size. This increase in the SPION size influences the water near it to diffuse in a single magnetic environment during the process of relaxation in a MR scanner. However, there is an optimal size limit, up to which this effect causes an increase in the relaxivity, which is termed as the static dephasing regime [136,137]. When the size of the SPIONs goes above the optimal limit, rapid dephasing of water protons takes place, even before the application of an echo pulse from the MR scanner [138]. This in turn decreases the relaxation ability of the contrasting agents. Thus, having SPIONs immobilized on a surface decreases the chances of clustering, thereby maintaining high relaxivity rates in comparison to free SPIONs. On the other hand, the encapsulation of SPIONs also provide a similar behavior of preventing the cluster formation. However, in this case, the encapsulation of SPIONs effectively reduces their interaction with the neighboring water molecules, thereby showing reduced contrasting abilities. Thus, the authors concluded that SPIONs immobilized on a polymer nanoparticle surface can serve as an efficient negative MRI contrast agent [139].

On the other hand, Jie et al. studied and proposed SPIONs encapsulated in a starch-octanoic acid (ST-OA) micelle as an efficient negative contrast agent which had the opportunity to also act as a drug delivery system thus making it a perfect targeted delivery system [140]. The ST-OA-SPIONs were in spherical shape with a size range of 105 ± 11.3 nm. This size limit falls well within the static dephasing regime, as discussed before. The size of the formed micelle increased to maximum of 170 nm upon encapsulating 10% doxorubicin (DOX). This system was used to study the in vivo magnetic property using a clinical MRI scanner with increasing Fe_3_O_4_ concentration. The experiment showed that, with increase in the concentration of Fe_3_O_4_, there was a linear decrease in the relaxation rate, suggesting that the formed SPIONs could serve as an efficient negative contrasting agent.

Another purpose of using Fe_3_O_4_ was to utilize its magnetic properties for targeting cancer cells. These SPIONs, when kept in an external magnetic field, are quickly accumulated at the site of application. Once the magnetic field is removed, the ST-OA-SPIONs showed a very good redispersion, which is an important aspect for obtaining a high-resolution image. The in vivo testing of the magnetic SPIONs showed that upon the application of a magnetic field at the tumor site, the ST-OA-SPIONs quickly accumulated in the tumor and were easily taken up by the tumor cells. This was monitored by using a fluorescence live animal imaging system and a fluorescent DOX that is released upon internalization of the micelle by the tumor cells. In addition, the fluorescence intensity was high due to the long circulation time of the ST-OA-SPION/DOX system. The presence of weak fluorescence, even 10 h after the accumulation of the micelles and removal of the external magnetic field, served as an additional benefit with regards to furnishing the contrast. Thus, such a magnetic SPION system could serve as an efficient targeted drug delivery system whilst behaving as an excellent contrasting agent [141].

Zhao et al. studied the use of Fe_3_O_4_ nanoclusters (NCs) coated with polyelectrolyte layers that contain poly (allylamine hydrochloride) (PAH) and poly (sodium 4-styrenesulfonate) (PSS), as an efficient contrasting agent. The Fe_3_O_4_ NCs were prepared by the microemulsion self-assembly method with 15 nm Fe_3_O_4_ NPs in a solid-in-oil-in-water (S/O/W) emulsion [142]. This process resulted in the formation of spherical nanoclusters of Fe_3_O_4_ NPs with an average dimension of 57 nm. The polyelectrolyte layers provide an alternatingly charged layer which could help in its drug carrying ability, thus providing an additional function to the nanoclusters, in addition to acting as an efficient contrasting agent. The hybrid nanostructure was subjected to in vitro analysis in MRI at 11.7 T with human lung cancer cells (A549) as the target. The transverse relaxation rates of (1/*T*_2_) were found to increase upon the addition of increasing concentrations of the formed nanoclusters. This behavior was rationalized by the agglomerated magnetic Fe_3_O_4_ core of the nanocluster [143,144], thus making the hybrid nanocluster a possible alternative to commercially available MRI contrasting agents [145].

Similarly, conjugated structures of IONPs have been developed for multi-functionality along with contrasting property. One such study was performed by Gonzalez-Rodriguez R et al.; GO-Fe_3_O_4_ NPs were developed to achieve cancer cell targeted drug delivery, assisted with MRI-based tracking to monitor the drug delivery. The average particle size of these NPs was 260 nm, which was achieved through the ultrasonication of the GO flakes [146]. The NP conjugates were subjected to in vitro testing on three different cancer cell types: HEK-293, HeLa and MCF-7. The MRI study revealed that the GO-Fe_3_O_4_ conjugate had an improved relaxivity ratio (r_2_/r_1_) over unconjugated Fe_3_O_4_ (here, r_1_ refers to the transverse T_1_ weightage and r_2_ refers to the T_2_ weightage in MRI). This phenomenon was rationalized by the presence of a GO layer, which prevents Fe_3_O_4_ interacting with the neighboring water molecules. This reduces the r_1_, along with a simultaneous increase in the formation of Fe_3_O_4_ nanoclusters at the GO surface that increases r_2_. Apart from this, the formed conjugate magnetic nanoparticles have very low coercivity levels, thus helping in easy demagnetization. With such benefits, the GO- Fe_3_O_4_ NPs were considered as a potential alternative to many commercially available negative contrasting agents [147].

In the pursuit of developing nanoparticle-based negative contrasts, C. Han et al. synthesized magnetic-fluorescent iron oxide-carbon hybrid nanomaterials (MCNP) that showed promising results compared to normal IONPs as a negative contrasting agent. These MCNPs were compared with IONPs that were prepared by the one-pot method. Both samples were tested in a human magnetic resonance scanner at a magnetic field of 3 T. The study revealed that the MCNPs showed a high degree of reduction in the transverse relaxation time and resulted in a significant reduction in the T_2_ weighted signal intensity at low concentrations of iron. Such a behavior exhibited an improved relaxivity ratio in comparison with pure IONPs (r_2_/r_1_ for MCNP = 8.74, while that of IONP = 2.08), thereby acting as an efficient negative contrasting agent. In addition to acting as a magnetic contrast agent, the hybrid MCNP system exhibited excitation wavelength dependent fluorescence. This was rationalized by the wide particle size distribution (21 nm–36 nm) in comparison to pure IONPs. In addition, the fluorescence lifetime exhibited by these MCNPs (6.23 ns) was considerably higher than the commercially available organic dyes (1–5 ns) and the inherent cell fluorescence exhibited by the proteins in the cell (1.5–4ns). Thus, the hybrid MCNPs could be used to effectively target multiple tissue types and help visualize them with ease [148].

All the above-mentioned studies explained the use of Fe_3_O_4_ in the form of SPIONs as a MRI contrast agent. While Fe_3_O_4_ SPIONs are the most widely studied contrast agents, other forms of iron oxide also show such properties. One such study was conducted by Kolesnichenko, V., G. Goloverda, et al., in which they used maghemite (γ-Fe_2_O_3_) as a potent MRI contrasting agent. γ-Fe_2_O_3_ and Fe_3_O_4_ nanoparticles were prepared in varying sizes (3.2 nm, 4.8 nm, 7.5 nm) to compare their relaxation ratios. The magnetic properties of the maghemite nanoparticles were studied, revealing ~20% less saturation magnetization values than Fe_3_O_4_. However, this did not affect its relaxation ratio. The γ-Fe_2_O_3_ NPs showed very similar relaxation ratio values as that of Fe_3_O_4_ across all sizes_._ The relaxation ratios were also compared to the commercially available MRI contrasting agent, gadolinium diethylenetriamine penta-acetic acid (Gd-DTPA). From this study, it was shown that γ-Fe_2_O_3_ NPs had better relaxation than Gd-DTPA. This is a great advantage considering the toxicity levels of Gd [149,150]. As most of the gadolinium derivatives are large complexes, they tend to accumulate in the body for longer time periods and can lead to many kidney and liver related disorders, which makes iron oxide nanoparticles a potent replacement. On the other hand, among the Fe^3+^ and Fe^2+^ oxidation states, Fe^3+^ is usually preferred due to its saturation magnetization value. Although Fe^2+^ has ~20% smaller magnetization values, it is relatively more stable than Fe^3+^. This means that there are far less toxicity effects in the body than with Fe_3_O_4_ NPs, as reactive nanoparticles could effectively create ROS in the body, which could lead to cell death and may alter many cellular pathways, thereby causing complexities. Thus, γ-Fe_2_O_3_ NPs could serve as a potential replacement to the frequently experimented Fe_3_O_4_ NPs [151].

Sometimes, these IONPs are tagged with radioisotopes to increase the precision in diagnosis with Positron Emission Tomography (PET) MRI screening. The contrast between the normal and diseased cells is possible by selective binding of the IONPs depending on the type of molecule coated on it.

### 6.2. Iron Oxide Nanoparticles in Immuno-Toxicity and Cell Toxicity

Immuno-toxicity, or cell toxicity, refers to chemically lysing a cell through the production of reactive oxygen species (ROS) that degrade the cell. Nanoparticles are an efficient system to produce ROS in the body. This is because nanoparticles (NPs) have a large surface to volume ratio, making them more reactive to the neighboring proteins, lipids, and other biomolecules, thereby disrupting their conformation and, in turn, hindering their molecular function. In addition, NPs have a large number of reactive sites due to their surface charge, crystallinity, and high surface area, which could lead to the production of ROS by reacting with surrounding molecules, which could in turn have a multitude of effects in the functioning of a cell. This property is used in the treatment of cancer by selectively delivering NPs to cancer cells which could cause cytotoxicity to lead the cancer cells to cell death, thus efficiently and selectively removing cancer cells from the body.

Ya-Na Wu et al. used the magnetic property of elemental IONPs for cancer treatment. Once positioned within a cell, they generate ROS that is caused by self-oxidation and surface interactions. The produced ROS then reacts with multiple cellular pathways and cell organelles such as the mitochondria. The ROS interacts with the mitochondrial membrane and destroys its potential gradient [152]. As the potential gradient of mitochondria is lost, its membrane molecules bind to apoptosis cofactors in the cytoplasm, ultimately leading to cell death by apoptosis. The disintegration of mitochondria also causes the activation of the necrotic pathways, thereby leading to cell death by necrosis. This property of Fe nanoparticles was used to selectively destroy cancer cells. As mentioned before, as Fe NPs have high reactivity, this process could only last for a couple of hours at maximum. In such a case, not all cancer cells are completely damaged; this could cause many more complexities, such as increased proliferation rates and metastasis. To avoid this and efficiently kill all the cancer cells, a gold coating was given to the IONPs to sustain the self-oxidation process, thereby causing slow and long-term effects. The process of the selective removal of cancer cells comes from the fact that cancer cells cannot recover from damaged mitochondria while the healthy cells can. In healthy cells, the damaged mitochondria are removed by sequestration and auto-phagocytosis, thereby preventing the downstream activation of the apoptotic and necrotic pathways. This natural difference between a healthy tissue and a cancerous tissue can be used to selectively remove the cancer cells, although the Fe NPs attack all cells in the vicinity, as shown in Figure 10 [153].

IONPs of varying shapes and sizes stimulate diverse immune responses in the body. In vivo studies were carried out in animal models by a single injection for a period of 13 weeks. The injected IONPs had multiple effects on the immune system and, in particular, the liver. IONPs with a 10 nm dimension and a spherical morphology primarily accumulated in the liver and spleen, and later penetrated other organs such as the brain and uterus. IONPs of higher dimensions were mostly accumulated in the kidneys. Apart from this, the NPs had affected the expression of certain cell surface markers and proteins such as CD40 and major histocompatibility complex (MHC) class II molecules. They also elevated the expression of neutrophils and suppressed the number of lymphocytes in the blood. Thus, IONPs can be used to induce immune toxicity in liver, spleen and thymus, predominantly [154].

Rakesh M. Patil et al., performed an extensive study on the effects of SPIONs, in vitro and in vivo, at various levels and have extended this cytotoxic property of SPIONs to therapeutic applications such as the targeted removal of cancer cells. As part of the in vitro studies, multiple cell lines from different tissue types were considered and subjected to a list of toxicity assay studies, including MTT [155,156,157], LDH leakage [158], trypan blue [159], and most frequently, comet assay [160]. These assays are used to evaluate the state of the cells based on staining in the different tests. The authors also proposed a set of possible SPION interactions at the cellular level. The SPIONs enter the cells through modes such as passive diffusion, receptor mediated endocytosis, clathrin mediated endocytosis, majorly. Once inside the cell, these SPIONs interact with various cell organelles such as mitochondria, lysosomes, and the nucleus. SPIONs, when bound to the mitochondria, cause damage to the mitochondrial membrane, thereby causing the cell to undergo stress due to the lack of ATPs. This stress, along with disruption of the mitochondrial membrane, causes the rise of ROS in the cytoplasm of the cell. ROS in turn causes damage to various cell organelles, cellular proteins such as metabolic enzymes, cytoskeletal proteins, DNA replication and repair mechanisms and even the DNA. Such a magnitude of damage to the cell drives the cell to its cell death. Apart from this, SPIONs also enter the nucleus, causing changes to the DNA structure, such as double strand breaks, point mutation, gene aberration and other mutations, thus destroying the integrity of the DNA. The damage to the DNA leads to a sequence of downstream reactions that lead to a lack of proteins and other necessary biomolecules, ultimately leading to cell death due to apoptosis.

On the other hand, the SPIONs can also damage the plasma membrane by undergoing redox interactions with the proteins of the plasma membrane. A collective effort from several SPIONs can cause disruption of the plasma membrane leading to necrosis [161]. In addition, the side effect of necrosis is that the spilling out of the cellular contents in the ECM of the cells causes further reactions and damage to the cells located in the tissue. Thus, reactive SPIONs can cause immense cytotoxic effects, leading to tissue and organ level damage.

SPIONs are used in biomedical applications for its diagnosis and therapeutic effects. Thus, the use of SPIONs will result in their accumulation in the targeted organ or tissue. The increased accumulation in the target tissue, although beneficial in terms of therapeutic and diagnostic applications, may lead to unwanted interactions and cause cellular damage by the various methods, as discussed before. Such a degree of cellular damage may also elicit the immune system to react against the accumulated nanoparticles, leading to the accumulation of large volumes of WBCs at the site of inflammation and leading to further complications. Moreover, the accumulation of large amounts of SPIONs may lead to their aggregation. Such aggregated structures result in micro- and milli-meter sized structures which cannot be processed/removed by the WBCs by phagocytosis. This leads to the formation of giant cells and results in frustrated phagocytosis, which can in turn elevate the inflammatory reactions at that site. Owing to such complexities, the use of SPIONs is conducted only in fewer concentrations to prevent the occurrences of such reactions [162].

Another group has worked on the cytotoxic effects of α-Fe_2_O_3_ NPs. In this study, human breast epithelial cells (MCF-7) were used to study the cytotoxic effects. The synthesized α-Fe_2_O_3_ NPs were analyzed to be in the dimensions of 19 nm. These NPs were added to the cells in varying concentrations of 25, 50, 100, 200 and 400 μg/mL and incubated for over 24 h to study the degree of cytotoxicity using a MTT assay. Based on the MTT assay, it was found that the cytotoxicity effect of the NPs was dose dependent, starting form 25 μg/mL, with a cell viability of 90%, to 400 μg/mL with a cell viability of 36%. These results were similar to other previously reported studies [163] and were confirmed using a trypan blue assay. An AO/PI staining assay was then performed to understand the mode of cell death. Both cytoplasm and nuclear staining revealed that the cell death was caused by the induction of apoptosis. These results were also confirmed using a flow cytometer to isolate the dead and live cells. The results from the flow cytometer also showed an increase in the number of apoptotic cells in a dose dependent manner. Thus, such a system could be employed in breast and other epithelial cell origin-based carcinoma cells to cause the apoptosis-based elimination of cancer cells upon the administration of sufficient concentrations of α-Fe_2_O_3_ NPs to the tumor site [164].

### 6.3. Iron Oxide Nanoparticles in Therapeutic Applications

Nanoparticles are used to provide better medical therapeutics, such as targeted drug delivery systems and hyperthermia, to name a few, due to its efficiency and targeted action reducing many side effects by avoiding unwanted damage of normal tissue.

In cases of hyperthermia, the SPIONs are used to generate heat at the tumor site and thus cause damage to the cancerous cells. This is possible by subjecting the cells with internalized magnetic nanoparticles to alternating magnetic fields, which could lead to the continuous movement of the NPs causing enough vibration energy to produce heat in the vicinity [165,166]. This heat could subject the cell to stress, leading to the increased production of ROS, followed by the degradation of cell organelles and damage to the nucleus and the DNA within. All these events lead the cell to apoptosis, thereby selectively eliminating the cancerous cell from the body without any side effects, which could not be possible in cases of high dosage of chemotherapeutic drugs. Thus, NPs can be stimulated to produce ROS locally to kill a cancer cells [167].

SPIONs coupled with hyperthermia techniques are used to kill the tumor cells via heat energy. The advantage of using SPIONs is that they are small, uniform, and are biocompatible when coated with specific polymers. This causes them to specifically bind to the tumor cells and subject them to a thermal shock, thereby killing them [168].

Dextran-stabilized IONPs (DIONP) were used for hyperthermia treatment. This composite was analyzed for its immunological response as even a slight effect in the iron content of the body might trigger an immune reaction. In this study, the DIONPs were confirmed to have immune stimulating levels well within the threshold, starting from platelet aggregation, lymphocyte stimulation mainly, thus classifying the drug as biocompatible and safe to use for hyperthermia treatments [169].

Wang et al. used dextran-stabilized IONPs to elevate the osteogenic differentiation of the human bone-derived mesenchymal stem cells (hBMSC). The formed IONPs were of core-shell architecture with a γ-Fe_2_O_3_ core and a polyglucose-sorbitol-carboxymethylether (PSC) shell. Here, the PSC was modified with dextran, which acts as a biocompatible stabilizer of the IONPs [170]. This process also prevents the leaching out of free iron. The cellular uptake of these IONPs was evaluated using the phenanthroline spectrophotometry assay by quantifying the iron content in the cell lysate after 24–72 h. This study showed that the IONPs were taken up by a time- and dose-dependent fashion. This result was confirmed using the Perl’s blue staining assay. The cell viability study showed that the IONPs had toxic effects with an increasing dosage and incubation time. The IONPs showed toxic effects even at 100 and 300 µg/mL when incubated for 24 h that were reflected in the viability assay, wherein it decreased by ~6% and ~8%, respectively.

The in vitro studies showed that there was an increase in the alkaline phosphatase (ALP) in presence of the IONPs, responsible for promoting osteogenesis [171,172,173]. However, this trend saturated after 100 µg/mL concentration of IONPs. The ALP activity was also found to be time dependent. The ALP activity showed a significant rise after 14 days post the introduction of IONPs and was compared with a standard osteogenesis inducing supplement (OS). The comparison revealed that both the IONPs and OS showed similar amounts of osteogenic differentiation.

After this, ARS staining was performed to check the effect of IONPs on the maturation of the IONPs. This study showed that the IONPs promoted the mineralization process, which was identified as mineralized nodules that were stained red when the cells were subjected to 100 and 300 µg/mL concentrations of IONPs. This was also compared to the effect of OS on the mineralization process. The mineralization process and an approximately 20% growth from the 14th to the 21st day of culture served as an identification of the transformation of the MSCs to mature osteoblasts. Throughout the growth period, the cells were monitored for their morphological changes. The growth pattern and changes in the shape of the cells during the process of differentiation, mineralization and maturation were similar in both IONPs treated and OS treated.

To understand the effect of IONPs in the differentiation of MSCs to mature osteoblasts at the gene level, certain osteogenic biomarkers were studied for their concentration in the culture medium. Markers such as FOXO1 [174], BMP2 [175,176,177,178,179] RUNX2 [180,181,182] and ALPL were tested for their mRNA expression levels using the Q-PCR technique. The results from the Q-PCR showed that these markers were overexpressed by a factor of 4–12 folds upon the administration of the IONPs to the in vitro culture. This was also supported by the increase in the respective gene expression. An in-depth study revealed the role of IONP in the cellular pathways. There was a noted enrichment in the MAPK pathway. This was substantiated by a set of 55 differentially expressed genes. The MAPK pathway plays an important role in cell proliferation and differentiation [183]. Thus, alterations to such pathways effect the accelerated differentiation of MSC to mature osteoblasts. Thus, IONPs were used in promoting osteogenesis, which serves as a bone repair mechanism [184].

Hybrid gold-IONPs were used due to their multi-functionality in cancer treatments and diagnosis. Fe_3_O_4_ NPs were prepared and coated with PEI, followed by gold and PEG. The polyethylene glycol (PEG) coat makes the NPs biocompatible [185]. This biocompatibility was tested in a cell viability experiment (trypan blue exclusion assay) with A375M cells. With the concentration of 25 µg/mL and an incubation period of seven days, there was no significant reduction in the viability of the cells cultured in vitro. These IONPs were then used in tandem with a laser for targeted hyperthermia treatments. The NPs were irradiated with a 532 nm laser, as it was close to its excitation wavelength. To check the spread of the heat generated, an additional thermocouple placed 14 mm apart from the one near the focal point of the laser was used. With varying irradiation periods (20, 40, 90 s), there was no significant heat measured at the second thermocouple, indicating that the laser irradiation was localized and did not cause any collateral damage. This was followed by the study of different concentrations of IONPs, each irradiated for 20, 40 and 90 s.

It was observed that lower concentrations of IONPs did significant increase the heat generated, with no clear increase in the irradiation time, while higher concentrations showed a drastic increase in temperature. The maximum temperature change recorded was 304 K with 50 µg/mL concentration for 90 s of irradiation time. In addition, there was no significant increase in the temperature in areas devoid of, or with a smaller, concentration of the IONPs in comparison to the areas that had high concentrations of IONPs.

To check the free IONP toxicity, these NPs were subjected to cell viability studies. The hybrid nanoparticles did not show a high degree of cytotoxicity on the A375M cells when exposed for seven days in an in vitro culture. The experiment showed a maximum toxicity of 20% decrease in cell viability in presence of NPs after seven days of incubation. This was significantly less than that of the free IONPs. The reduction in toxicity was attributed to the gold coating over the IONP core, which increases the size of the NP and also prevents the enzymatic degradation of iron from the core which could result in free radical production in the cytoplasm; this can in turn cause cellular damage, thereby causing cell death [186]. This can therefore be used to target the cancer cells based on the cell surface markers of the cancer cells. In addition, the property can be used for targeted drug administration by laser activation. As the IONPs have a magnetic property, they can also be used as contrasting agents in MRI for the detection of cancer cells. This was also tested, and it was found that these hybrid nanoparticles have increased relaxation ratios (~555.55), thus making them a potential negative contrast agent for MRI [187].

Magnetic iron oxide nanoparticles were used in in vitro studies of hyperthermia on tumor cells in agarose gel. It was found that the nanoparticles embedded in the gel had a better effect in raising the temperature to the threshold of 316 K, rather than in bulk solutions [188].

Although this seems an efficient method of eliminating the cancerous cells from the body, hyperthermia as a standalone process is not effective. This is because the treatment procedures should be short, as long exposure can lead to the damage of the neighboring healthy cells. Moreover, as the SPIONs accumulate at the tumor site, the cumulative heat produced could also result in the damage of cells in the vicinity. Thus, hyperthermia is used as an adjuvant therapy to common treatment procedures for cancer, such as chemotherapy and radiotherapy.

Kebede et al. used FeO NPs in combination with chitosan for the oral delivery of insulin. For this, the FeO NPs were synthesized using the high-power laser ablation method, followed by immediate thermal quenching, and were formed in a chitosan medium. This is because chitosan has a good chelation property with metal ions and metal oxides separating the metal ions from each other [189]. Thus, chitosan acts as an efficient stabilizer to maintain the FeO phase NP. Once the unstable oxide was formed and stabilized, these NPs were tested for their drug loading and release properties. Three groups of rats were chosen for this study. The first group of rats had sub-diabetic conditions, the second group had mild-diabetic conditions and the third group of rats had severe-diabetic conditions. All of these rats were artificially induced with diabetes by various clinical procedures. These rats were then divided into four sections per group to test the effect of free insulin, insulin in IONP, insulin in IONP coated with chitosan and a control group that was administered with only distilled water. The rats were administered with the drug and subjected to a glucose tolerance test at 0 h, 1 h and 2 h after administration.

From this study, it was found that the sub-diabetic rats had around 34% reduction in the blood glucose levels on administration, while there was no significant decrease in the blood glucose levels in the animals administered with free insulin and insulin in IONPs showed only a decrease of ~15%.

This reduction in the blood glucose value then dropped to 22% at 2 h after administration. In the case of the mild-diabetic rats, the insulin loaded in IONP with chitosan showed a decrease of 22% in the blood glucose after 1 h, that later decreased, while the free insulin and insulin loaded IONPs had no significant decrease. With the severe-diabetic rats, the decrease in blood glucose upon administration of insulin loaded IONPs with chitosan was about 51% and that of insulin loaded IONPs were ~40% after 1 h. However, even in this case, the free insulin had no significant effect in reducing the blood glucose. These values remained after 2 h, suggesting the efficacy of the treatment. All of these values of reduced blood glucose levels were measured relative to the control group. During the experiment, a slight increase in the blood glucose levels were observed. This was rationalized to the stress caused to the animal by the process of blood collection. The stress caused can increase the secretion of glucagon which is ultimately converted to glucose in the blood [190]. The improved activity of the insulin loaded IONPs (both with and without chitosan) can be rationalized based on the availability of insulin within and outside the cell. In diabetic conditions, cells do not take up the insulin produced due to insensitive insulin receptors. This leads to diabetes type 2, where even in the presence of enough concentration of insulin in the blood, there is no reduction in the blood glucose levels. Therefore, even with the administration of additional insulin, there is an insignificant effect.

On the other hand, insulin loaded IONPs are taken up by the cells. The insulin release takes place only when the IONPs are inside the cell, bypassing the involvement of the dysfunctional insulin receptor [191]. When the insulin is already inside the cell, it induces the cellular pathways to reduce the blood glucose levels efficiently [192]. Regarding the differential activity of insulin loaded IONPs with chitosan and without a chitosan coat, the presence of chitosan makes the insulin form only a weak electrostatic interaction and weak hydrogen bonding with the NP surface, which does not occur in the case of IONPs without chitosan coating [193]. This altered interaction condition is the reason behind the faster release of large amounts of insulin from the IONPs with chitosan, thereby effectively decreasing the blood glucose levels in the severe-diabetic rats. By such a mechanism, the composite had far better results in reducing the blood glucose levels than IONPs + insulin and pure insulin, making it a drug for diabetic patients [194].

IONPs coated with gold were used to suppress oral cancer and colorectal cancers as they induce the formation of reactive oxygen species which leads to cytotoxicity. In addition, the gold coated IONPs affect the functioning of mitochondria by decreasing its membrane potential. This leads to a cascade of reactions and ultimately necrosis.

IONPs are used in cancer treatments by encapsulating the drug within the IONPs and coupling it with suitable markers to achieve targeted drug delivery, thereby providing an efficient cure for it. IONPs were used as carriers for doxorubicin, an anticancer drug. IONPs of varying sizes were synthesized and tested for their drug delivery efficiency on HT29 cells. It was found that Dox-NPs had better efficiency than free Dox and, although the size of the nanoparticles did not affect the efficiency of drug delivery and IONPs of sizes 10 nm had better penetration into the cancer cells (HT29), the absorption of the drug was tested and confirmed with fluorescence spectroscopy [195].

### 6.4. Iron Oxide Nanoparticles in Biosensing Applications

Nanoparticles play a significant role in the field of biosensors. Apart from the requirement of the miniaturization of the sensing element, nanoparticles have been used for their remarkable properties, such as enhanced catalytic activity, rapid charge transfer between the electrolyte and the active surface of the working electrode, providing functional sites for binding of enzymes and maintaining the stability of the bound enzyme. Such properties of the nanoparticles furnish a wide range of advantages, such as improved sensitivity, enhanced electron transfer rate, prevent bilayer formation and provide stability to enzymes specific to the analyte. Hence, nanoparticles modified by electrodes furnish the detection of analytes even at femtomolar ranges [196]. All of these factors improve the precision of the fabricated biosensor to detect concentrations of a broad spectrum of biologically important molecules, such as cancer biomarkers (antigens [197], exosomes [198], miRNAs [196]), Adenosine triphosphate (ATP) [199], hormones such as dopamine [200,201], epinephrine [202], glucose [203], and uric acid [200] predominantly.

An enzymatic biosensor was developed for cholesterol detection by immobilizing cholesterol oxidase with micro-pine shaped IONP (α-Fe_2_O_3_). Although the developed sensor had an unsuitable LOD (limit of detection), the micro-pine shaped IONP structures had high sensitivity, which can be put into use in other sensing fields [204]. A lactase-based biosensor was fabricated for determining hydroquinone. In this work, it was found that the graphene-chitosan-Fe_2_O_3_ nanocomposite showed a better film-forming ability and good electrical conductivity, which was used thereafter in the sensing application [204].

Functionalized IONPs were used in bio-sensing applications as they exhibited biocompatibility and signal amplification. This property was used in the simultaneous detection of tumor markers in a multiplexed immunoassay for the early detection of cancer. One such work was the simultaneous detection of alpha-fetoprotein (AFP) and carcinoembryonic antigen (CEA) levels, utilizing metallic NPs coated with a recombinant apoferritin (rApo-M) as labels and dual-template magnetic MIPs (MMIPs) as capturing probes, which showed a wide (or lower) range of LOD [205].

Magnetite nanoparticles were used in an electrochemical immunosensor application for the detection of Vitamin D3. This was achieved by incorporating the NPs in a polyacrylonitrile nanofiber matrix, followed by functionalization with Vitamin D3 specific monoclonal antibodies. All of these layers were placed on an ITO (indium tin oxide) substrate and employed as a sensor. The fabricated sensor was reported to show improved biosensing properties, with LOD of 0.12 ng mL^−1^ and a detection range of 10–100 ng mL^−1^, as shown in Figure 11 [206].

Similar methodologies have been adopted in cholesterol biosensing, where Fe_3_O_4_ and α-Fe_2_O_3_ NPs were coupled with a cholesterol oxidase (ChOx) enzyme on an ITO substrate to obtain improved biosensing properties. The fabricated sensor was compared with other metal oxide choices for cholesterol biosensing and was found to have a comparably improved linear range (Fe_3_O_4_/ChOx/ITO—25–400 mg dL^−1^, α-Fe_2_O_3_/ChOx/ITO—50–400 mg dL^−1^) and sensitivity (Fe_3_O_4_/ChOx/ITO—193 nA mg^−1^ dL cm^−2^, α-Fe_2_O_3_/ChOx/ITO—50–400 nA mg^−1^ dL cm^−2^), without forgoing the specificity of the sensor owing to use of enzymatic methods. In addition, there was an increase in the reproducibility and shelf life of the sensor in comparison with the other metal nanoparticles [207].

IONPs were also used for enzymatic glucose sensing applications. γ-Fe_2_O_3_ nanoparticles were coupled with glucose oxidase and citric acid as the functionalizing agent by drop casting onto an ITO substrate and were employed as a glucose sensor. Due to the enhanced electron transfer rates, the sensor showed high sensitivity of 70.1 µA mM^−1^ cm^−2^ with a linear range of 1–8 mM, as observed from Figure 12 [208].

While the above-mentioned sensors employed the use of IONPs as the only nanointerface, IONPs coupled with other nanomaterials, such as CNTs, have been employed in biosensing applications. Fe_3_O_4_ NPs, along with multiwalled carbon nanotubes (MWCNTs), were used as a hybrid nanointerface for the detection of hydrogen peroxide (H_2_O_2_) in milk, thus assessing the quality of milk. The fabricated sensor was used for amperometric detection and the quantification of H_2_O_2_, which showed improved sensing properties compared to the existing biosensors, with a good detection range (1.2–21.6 µM), quick response time of less than 1 s and high sensitivity of 0.5732 nA µM^−1^cm^−2^, evident from Figure 13 [209].

In another study, Fe_2_O_3_ NPs functionalized with MWCNT were used as a nanointerface for the detection of Levodopa. This fabricated sensor was proposed to have improved linear range (0.3–8 µM) and good limit of detection (LOD–0.24 µM), evident from Figure 14. In addition, the proposed sensor architecture had a better specificity in the presence of interference from other analytes, even at high concentrations [210].

In a recent study, maghemite nanoparticles were synthesized through a green synthesis co-precipitation route by making use of Furostanol Saponin (FS), a bio-surfactant in Fenugreek seeds extract. Three different morphologies, namely nanospheres, nanowires and nanograsses, were obtained with the same crystal structure and magnetic properties but with varied size distribution. It was observed from their electrochemical studies that the maghemite nanowires exhibited superior sensing characteristics compared to the other two morphologies, in the form of discernable voltametric signals for the simultaneous detection of dopamine and uric acid, along with an increased oxidation current, which is generally a significant challenge in the design of biosensors performing the simultaneous detection of these two biomolecules. The DPV results exhibited a linear range of 0.15–75 μM for dopamine and 5 μM–0.15 mM for uric acid with the LOD being 150 nM and 5 μM, respectively.

Similarly, graphene oxide-loaded iron oxide hybrid nanointerface was used for the ultrasensitive detection of miRNA. This was facilitated by the use of ruthenium hexa-amine (III) chloride as an intermediate to facilitate the ultra-sensitivity, as represented in Figure 15. This sensor was employed in the detection of cancer-specific miRNA by the chronocoulometric method, thereby efficiently detecting cancer and the disease progression. This novel technique showed an exceptional LOD of 1.0 fM to 1.0 nM, which helps in the accurate detection of cancer miRNA. Along with this, the proposed sensor displayed a high sensitivity (10 cells) and good reproducibility (with %RSD < 5%, for *n* = 3), when employed in clinical testing of ovarian cancer cells [211].

Numerous interesting nanocomposites containing iron oxide nanoparticles have been synthesized and used for the electrochemical detection of a wide range of biomolecules, with huge relevance and relating to high prevalence diseases. For example, a sandwich type electrochemical biosensing platform was constructed out of graphene oxide/Prussian blue (GO/PB) as the probe and spiky Au@Fe_3_O_4_/MUC1 aptamer as the capture unit for detection of cancer-related MCF-7 exosomes, as shown in Figure 16. The chief highlights of this work are that it shows a very low limit of detection, ~ 80 particles/μL, and does not require any pre-processing in untreated serum samples. The MUC1 aptamer was considered, in this case, owing to its overexpression in MCF-7 exosomes; Au/Fe_3_O_4_ combination due to their excellent conductivity, catalytic property, and rapid electron transfer. They reported a linear range of 200–5 × 10^5^ particles/μL with a response time of 40 min, considered to be state-of-the-art in exosome detection relating to a cancer type [198].

Similarly, a paper-based disposable, label-free, conducting paper-based immune-electrochemical biosensor was developed to detect CEA, an important biomarker for various cancers, such as colon and rectum, prostate, lung, liver, thyroid, and ovarian. This biosensor made use of poly(3,4-ethylenedioxythiophene): poly(styrene sulphonate) (PEDOT:PSS) and nanostructured iron oxide nanocomposite, which was coated onto a Whatman filter paper. This sensor exhibited a linear range of 4–25 ng/mL, a sensitivity of 10.2 μA mL cm^2^/ng and a longer shelf life of 34 days, compared to similar works [197].

Further, the recent literature on cancer biomarkers have reported the effectiveness of considering miRNAs as reliable biomarkers for different stages of various cancers as they take part in important cellular processes, such as cell proliferation, cell cycle regulation, DNA repair and apoptosis [212]. Various miRNA detection strategies have been proposed, modelled [213] and reported [214,215,216].

One such example, out of the numerous reports, made use of the hybridization chain reaction (HCR) with Fe_3_O_4_@SiO_2_@AuNPs coated with hairpin cDNA as the capture element and HCR-Ru(phen)_3_^2+^ as the signal element, which is a long dsDNA that was obtained from HCR and subsequently embedded with Ru(phen)_3_^2+^ labels. The detection mechanism of this biosensor is as follows: in the presence of the target miRNA-126, the stem-loop structure of the cDNA present in the capture element opens, resulting in the formation of a partial dsDNA, which subsequently hybridizes with the signal element to form a complex on the magnetic glassy carbon electrode surface. Electrochemical Impedance Spectroscopy (EIS) and Electrochemiluminescence methods were employed for the detection and the biosensor exhibited a LOD of 2 fM.

Table 2 summarizes the different strategies used for electrochemical sensing of the broad spectrum of biomolecules along with their figure of merits.

### 6.5. Iron Oxide Nanoparticles as Anti-Bacterial Agents

IONPs exhibit bactericidal activity through the deactivation of the cellular functions of the bacteria. *Staphylococcus aureus* was treated with poly (vinyl alcohol)-coated IONPs in varying doses/concentrations. There was a significant decrease in the concentration of *S. aureus*, in concentrations as low as 3 mg mL^−1^. This is due to the action of IONPs as oxidative stress inducers, leading to the production of reactive oxygen species (ROS) that induce cell death by causing damage to the DNA and protein [217,218]. Similarly, IONPs treated with *Argemone mexicana* leaf extract had an enhanced bactericidal effect against *E. coli* [219] and IONPs had a significant influence in the reduction in the biofilm formation on the biomaterial surfaces [220].

## 7. Conclusions

The present review on iron oxide nanoparticles has provided an extensive discussion on the province of nanoparticles fashioned out of 14 polymorphs of iron oxides. Among the copious strategies and their corresponding techniques or synthesis methods for synthesizing the iron oxide nanoparticles, some of the stand-alone and widely used methods, such as coprecipitation, hydrothermal, sol-gel methods, were reviewed along with state-of-the art techniques such as sonochemical, electrochemical and plasma-based methods. In addition, the structural properties of the polymorphs of iron oxide nanoparticles synthesized to date have been conferred and have been correlated with the properties they manifest when used for variegated applications, most significantly in biology. Notwithstanding the properties, the phase transitions between the polymorphs of iron oxides have been reviewed, while associating its influence over the possible oxidation states of iron.

Looking back from here, it might seem that iron oxide nanoparticles have been exhaustively explored, synthesized, and found its use in the mainstream applications which gave rise to the current review. At the same time, presently, we have reported the discovery of new polymorphic forms of iron oxides such as the ζ-Fe_2_O_3_ phase. Hence, looking into the future, by considering the material aspect alone, we have presented a compelling scope to discover new forms and to engineer them for very specific applications, thereby creating a world that is made of this ubiquitous material. Considering the biological applications, the maneuvering has been tremendous. The diverse properties of iron oxide nanoparticles employed in the fields of imaging, sensing, diagnostics, and therapeutics have paved the way to more efficient disease diagnosis and cures, with the additional benefits of very minimal side effects. Propitious studies, such as on the structural and property characterizations of other less explored polymorphs of iron oxides and the better translation of structure to function among conventionally used iron oxide polymorphs are encouraged in the future for this ubiquitous metal oxide.

## Figures and Tables

**Figure 1 materials-16-00059-f001:**
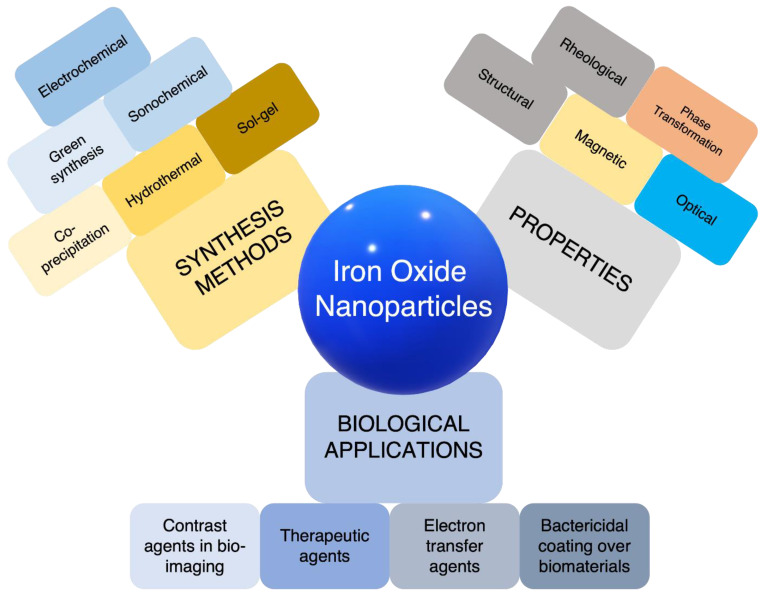
Overview of synthesis methods, properties and biological applications of iron oxide nanoparticles that have been extensively discussed in the current review.

**Figure 2 materials-16-00059-f002:**
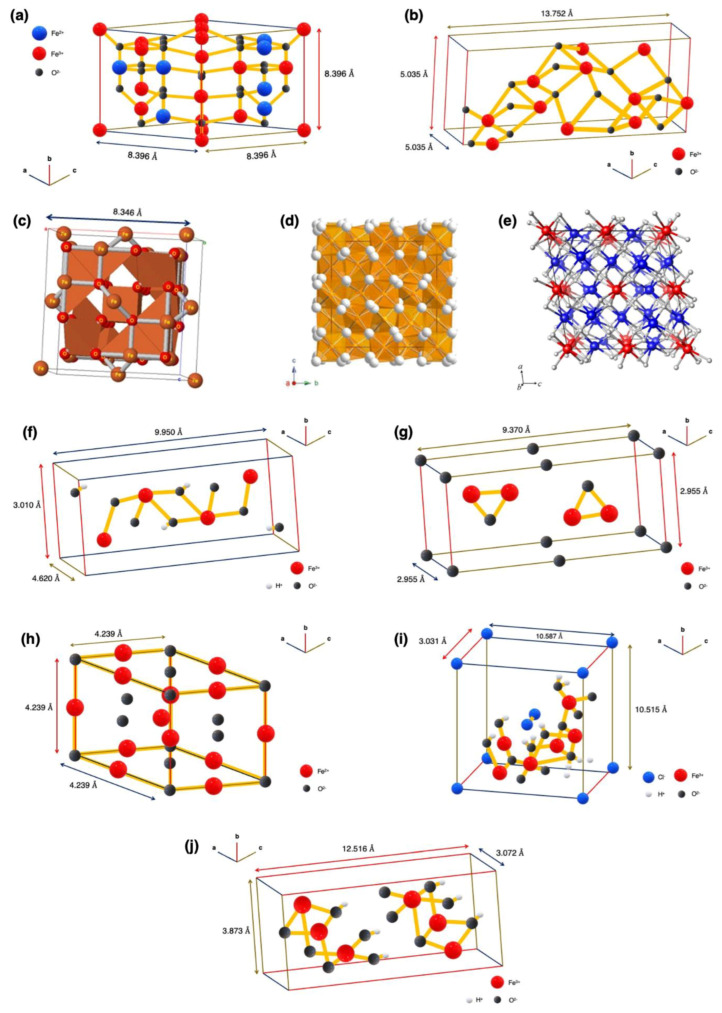
Crystal structures of different polymorphs of iron oxide (**a**) Magnetite (**b**) Hematite (**c**) Maghemite (**d**) β-Fe_2_O_3_ (Reprinted with permission from Ref. [43] Copyright 2013, American Chemical Society) (**e**) ζ -Fe_2_O_3_ [5] (**f**) Goethite (**g**) Ferrihydrite (**h**) Wüstite (**i**) Akaganeite (**j**) Lepidocrocite.

**Figure 3 materials-16-00059-f003:**
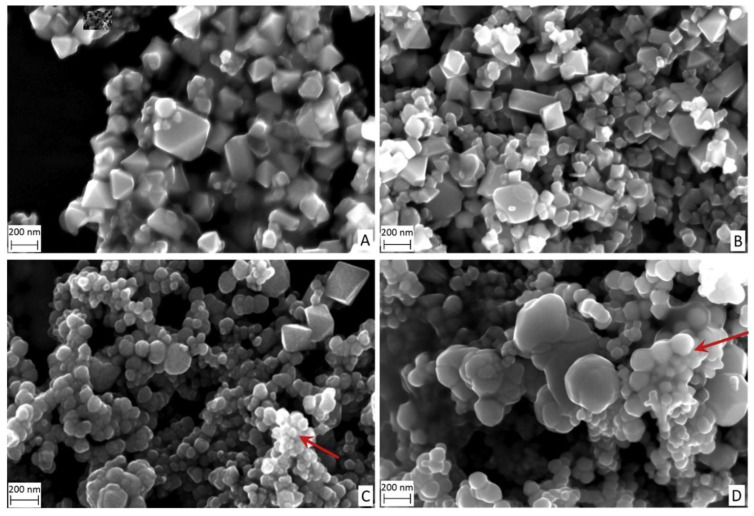
SEM images of magnetite nanoparticles synthesized at different *p* values (**A**) *p* = 0.57 (**B**) *p* = 0.56 (**C**) *p* = 0.21 (**D**) *p* = 0.16 (Reprinted with permission from Ref. [47] Copyright 2019 Elsevier).

**Figure 4 materials-16-00059-f004:**
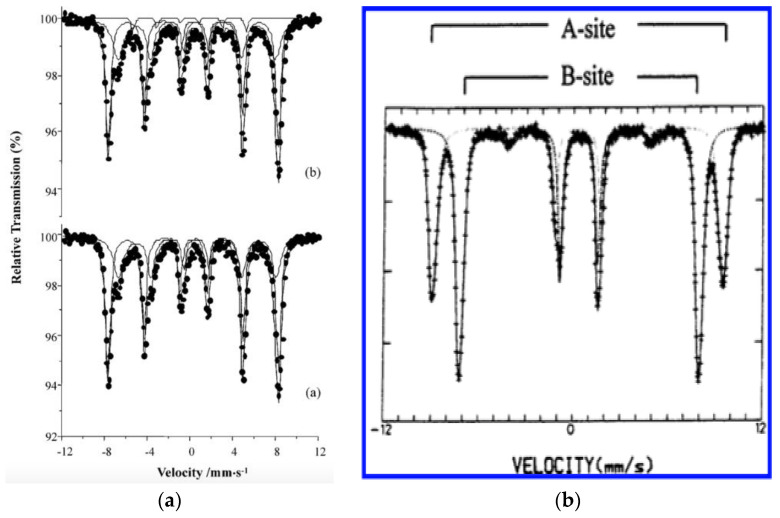
Mossbauer spectrum of (**a**) magnetite (Reprinted with permission from Ref. [21] Copyright 2008 Elsevier) (**b**) maghemite (Reprinted with permission from Ref. [49] Copyright 2002, American Chemical Society).

**Figure 5 materials-16-00059-f005:**
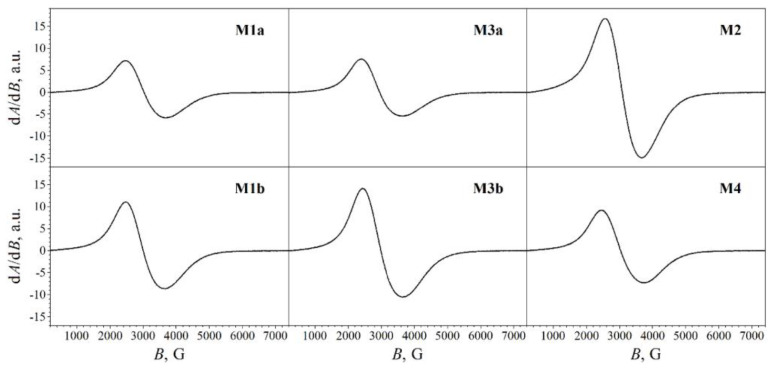
X-band Electron magnetic spectra of magnetite nanoparticles prepared by co-precipitation method (Reprinted with permission from Ref. [45] Copyright 2019 Elsevier).

**Figure 6 materials-16-00059-f006:**
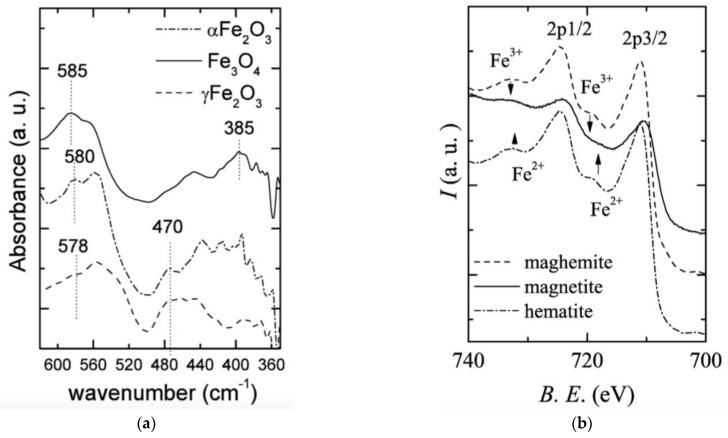
(**a**) FT-IR spectra for magnetite, maghemite and hematite nanoparticles (**b**) XPS spectra of Fe2*p* from magnetite, maghemite and hematite nanoparticles (Reprinted with permission from Ref. [43] Copyright 2017 Elsevier).

**Figure 7 materials-16-00059-f007:**
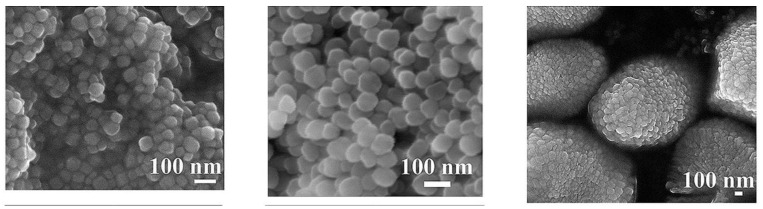
Morphological characterization of hematite nanoparticles that were synthesized using different concentrations of surfactants—S1, S2 and S3 (**left** to **right**) (Reprinted with permission from Ref. [55] Copyright 2019 Elsevier).

**Figure 8 materials-16-00059-f008:**
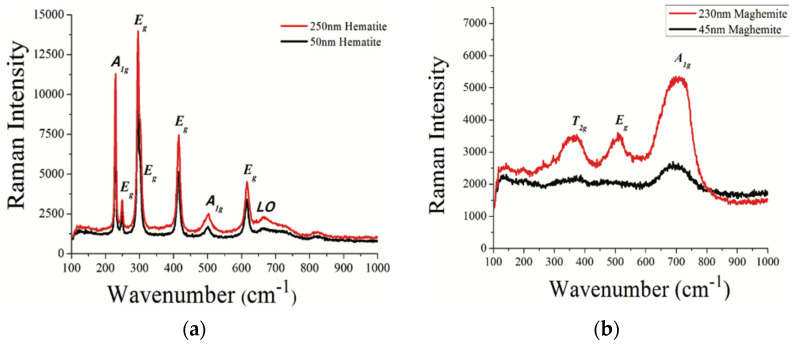
Raman spectra for (**a**) hematite (**b**) maghemite nanoparticles (Reprinted with permission from Ref. [64] Copyright 2010 American Chemical Society).

**Figure 9 materials-16-00059-f009:**
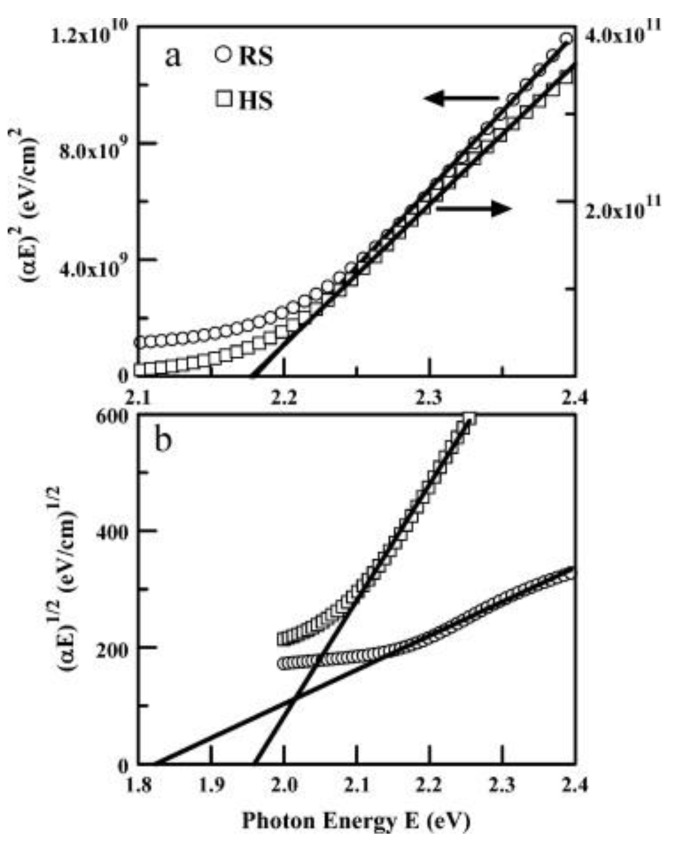
Tauc plots of hematite films deposited on unheated (RS) and heated (HS) substrates showing (**a**) direct band gaps and (**b**) indirect band gaps (Reprinted with permission from Ref. [108] Copyright 2012 Elsevier).

**Figure 10 materials-16-00059-f010:**
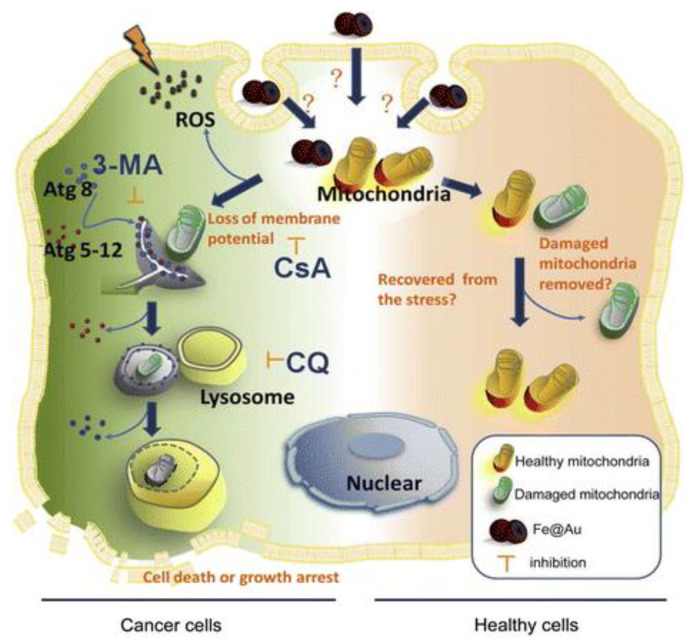
Illustration of mitochondria-mediated autophagy induced by Fe@Au composite material for cancer cell specific cytotoxicity (Reprinted with permission from Ref. [153] Copyright 2011 Elsevier).

**Figure 11 materials-16-00059-f011:**
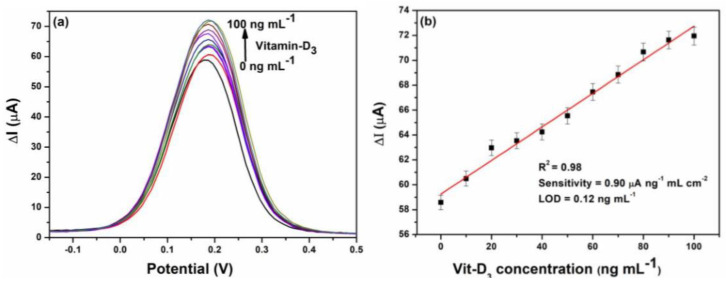
(**a**) DPV response of immunoelectrode (BSA/Anti-VD/Fe_3_O_4_-PANnFs/ITO) as a function of antigen concentrations (**b**) Calibration graph between current peak and antigen concentration (Reprinted with permission from Ref. [206] Copyright 2018 Elsevier).

**Figure 12 materials-16-00059-f012:**
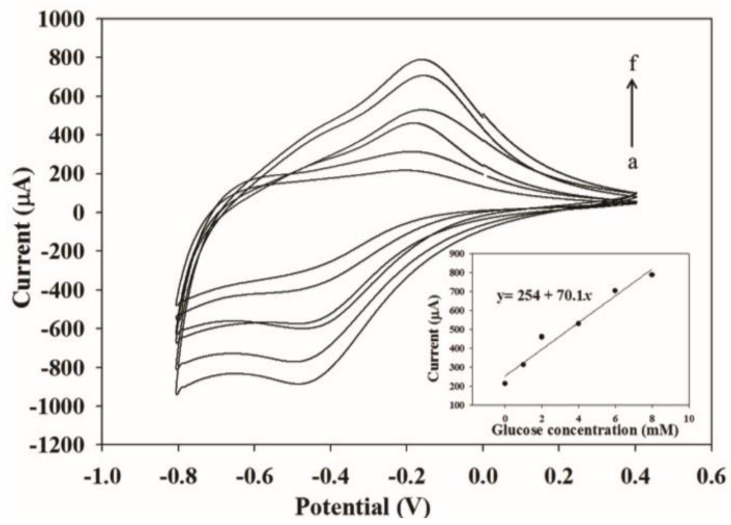
Cyclic Voltagramm plots of Nafion/GOx/CA-IONPs/ITO bioelectrode (**a**) in absence of glucose (**b**–**f**) in 1, 2, 4, 6 and 8 mM glucose into 0.1 M PBS (pH 7.0) at scan rate 100 mV/s. Inset: Calibration curve of current obtained as a function of glucose concentration (Reprinted with permission from Ref. [208] Copyright 2016 Elsevier).

**Figure 13 materials-16-00059-f013:**
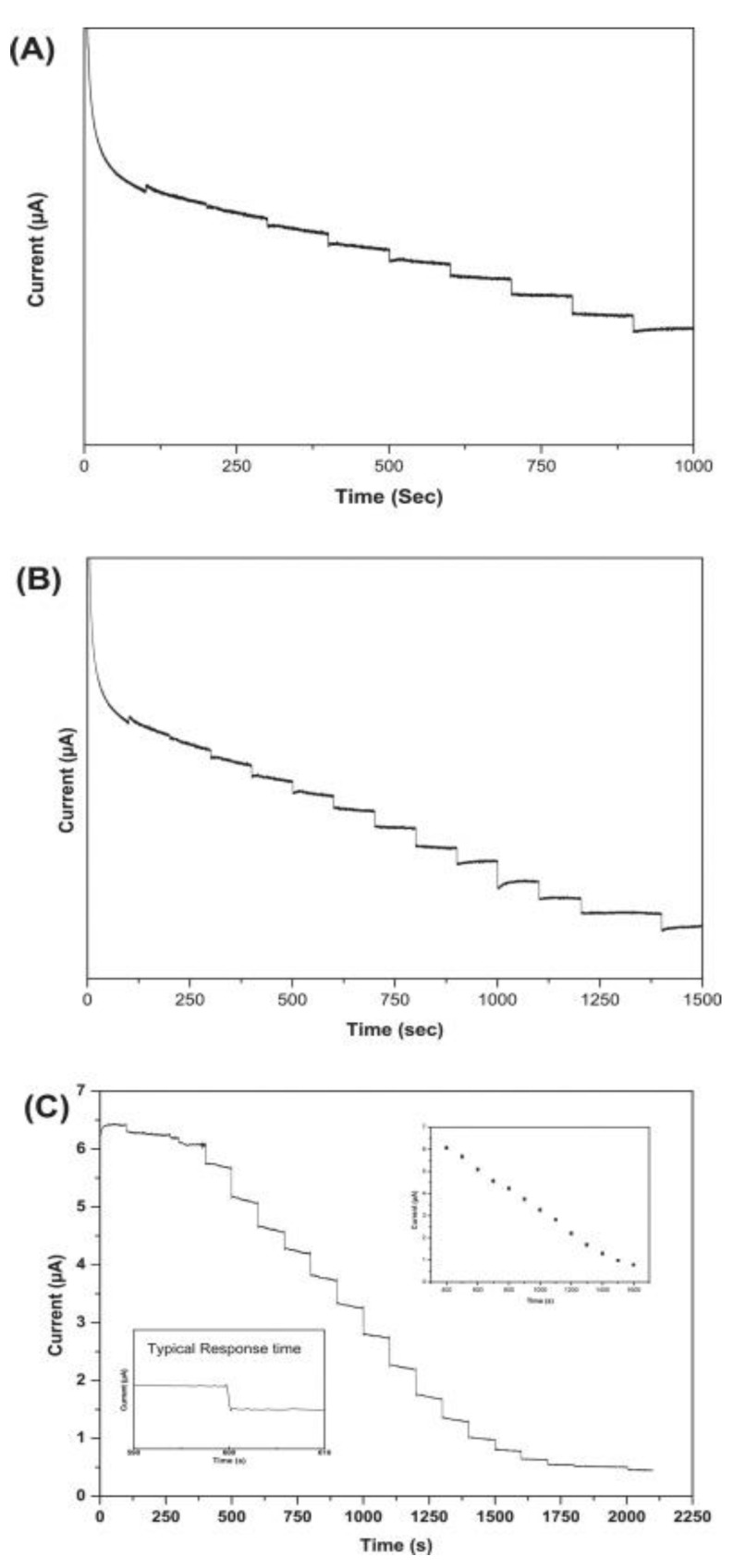
Amperometric response (**A**) CAT/Fe_3_O_4_/Au (**B**) CAT/CNT/Au (**C**) CAT/Fe_3_O_4_–CNT/Au hybrid electrode for successive addition of 1.2 M hydrogen peroxide, at a constant potential of −0.05 V using phosphate buffer solution (pH 7.4) as electrolyte (Reprinted with permission from Ref. [209] Copyright 2015 Elsevier).

**Figure 14 materials-16-00059-f014:**
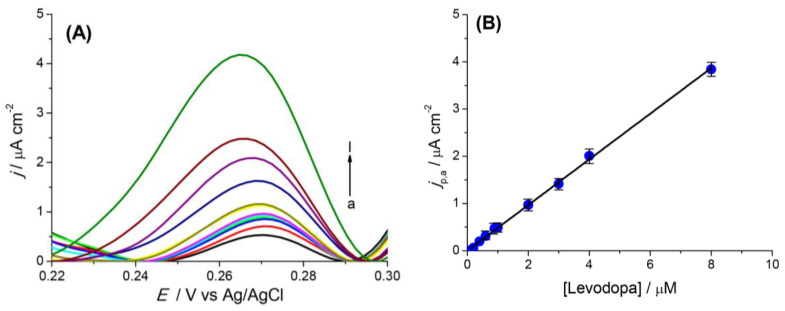
Differential pulse voltammograms (**A**) and calibration curve (**B**) (from 3 independent measurements at each concentration) for levodopa at GCE modified with (0.2% Fe_2_O_3_ 1% MWCNT), in 0.1 M PB solution, pH 6.0. Scan rate 4 mV/s; step potential 2 mV. Concentration of levodopa: (a) 0.1; (b) 0.2; (c) 0.4; (d) 0.6; (e) 0.8; (f) 0.9; (g) 1.0; (h); 2.0; (i) 3.0; (j) 4.0; (l) 8.0 mM (Reprinted with permission from Ref. [210] Copyright 2018 John Wiley and Sons).

**Figure 15 materials-16-00059-f015:**
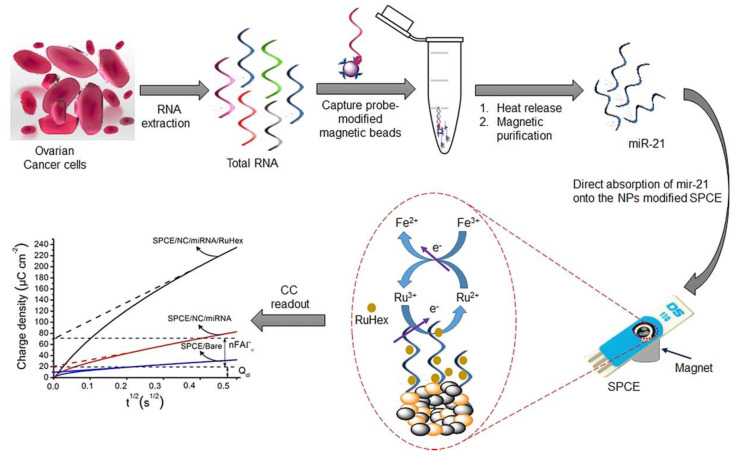
Schematic of the quantification of miRNA assay. Magnetically purified and separated miRNA from the extracted RNA sample pool were adsorbed directly on the magnetically bound GO/IO hybrid- modified SPCE. A significant electrocatalytic signal amplification was achieved via the chronocoulometric (CC) charge interrogation of target miRNA-bound [Ru(NH3)6]3+- [Fe(CN)6]3- electrocatalytic assay system (Reprinted with permission from Ref. [211] Copyright 2018 John Wiley and Sons).

**Figure 16 materials-16-00059-f016:**
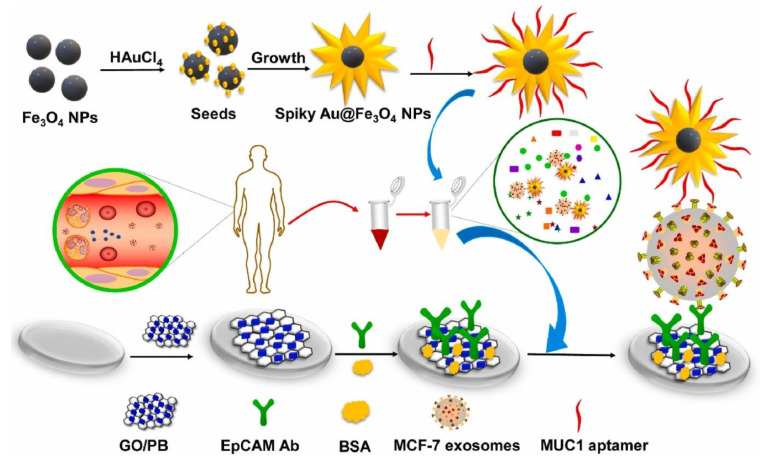
Schematic representation of the electrochemical biosensor developed for detection of MCF-7 exosomes along with the detection mechanism (Reprinted with permission from Ref. [198] Copyright 2022 Elsevier).

**Table 1 materials-16-00059-t001:** Structural, Magnetic and Phase Transforming properties of Iron Oxide Nanoparticles Phases.

*Iron Oxide Phases*	Chemical Formula	Oxidation State	Crystal Structure	Magnetic Properties	Influence on Oxidation States and Phase Transformations	References
*Magnetite*	Fe_3_O_4_	+2 and +3	Face centered Cubic (a = b = c = 0.8396 nm)	Ferrimagnetic	Higher oxidation rate to maghemite when left at room temperature as size of nanoparticles decrease	[6,70,101]
*Hematite*	α-Fe_2_O_3_	+3	Rhombohedral with hexagonal closed packing (a = 0.5035 nm, c = 1.3752 nm)	Antiferromagnetic below 955 K and paramagnetic above 955 K	Transformed to have increased oxygen vacancies to have enhanced electrochemical performance	[6,70,120,125]
*Maghemite*	γ-Fe_2_O_3_	+3	Allotropic form of magnetite Cubic (a = 0.8346 nm)	Ultra-small nanoparticles exhibiting super-paramagnetism	Transformed into hematite directly or indirectly with ϵ-Fe_2_O_3_ as intermediate	[6,70,71,120]
*Iron Oxide beta phase*	β-Fe_2_O_3_	+3	Body centered Cubic (a = 0.9393 nm)	Paramagnetic between 100–119 K and antiferromagnetic below 100 K	Directly transforms to hematite/magnetite or even ζ-Fe_2_O_3_ under high-pressure transformation but hollow nanoparticles transform into maghemite	[6,70,120]
*Iron Oxide eta phase*	ϵ-Fe_2_O_3_	+3	Orthorhombic (a = 0.5072 nm, b = 0.8736 nm, c = 0.9418 nm)	Two magnetic transitions 1. Paramagnetic to Meta ferromagnetic at 495 K 2. Meta magnetism below 110 K	Directly transforms to hematite	[70,78]
*Iron Oxide zeta phase*	ζ-Fe_2_O_3_	+3	Monoclinic (a = 0.9683 nm, b = 1.0000 nm, c = 0.8949 nm)	Transition from paramagnetic to antiferromagnetic state at 69 K	N.A.	[5]
*Goethite*	α-FeO(OH)	-	Orthorhombic (a = 0.995 nm, b = 0.301 nm, c = 0.462 nm)	Superparamagnetic	N.A.	[6,112]
*Ferrihydrite*	Fe_5_HO_8_.4H_2_O	-	Hexagonal (a = 0.5958 nm, c = 0.8965 nm)	Ferromagnetic—increase in property with increase in surface to volume ratio	N.A.	[6,97]
*Wüstite*	FeO	+2	Cubic (a = 0.4239 nm)	Antiferromagnetic	N.A.	[6,107]
*Akaganeite*	β-FeO(OH)	-	Monoclinic (a = 1.0561 nm, b = 3.031 nm, c = 1.0483 nm)	Exhibits magnetic birefringence when coupled with polysaccharide solution Particle size brings anomalies	N.A.	[6,98,99]
*Lepidocrocite*	γ-FeO(OH)	-	Orthorhombic (a = 0.388 nm, b = 1.254 nm, c = 0.307 nm)	Antiferromagnetic	N.A.	[6,100]

**Table 2 materials-16-00059-t002:** Summary of different electrochemical strategies used for biosensing of various biomolecules.

*Biosensor Components*	Detection Method	Electrochemical Method Employed	Biomarker Detected	Linear Range	Detection Limit (LOD)	References
*Recombinant apoferritin-encoded Fe_3_O_4_ nanoparticles + dual-template magnetic molecularly imprinted polymers*	Immuno-sensing	Square wave voltammetry	Simultaneous detection of AFP and CEA	0.001–5 ng/mL	AFP—0.3 pg/mL CEA—0.35 pg/mL	[205]
*Magnetite nanoparticles + polyacrylonitrile nanofibers*	Immuno-sensing	Differential Pulse Voltammetry	Vitamin-D_3_	10–100 ng/mL	0.12 ng/mL	[206]
*Magnetite Nanoparticles passivated with carbon shell*	Enzymatic	Cyclic Voltammetry and Impedance Spectroscopy	Cholesterol	25–400 mg/dL	Not reported	[207]
*Citric acid capped maghemite nanoparticles + Nafion*	Enzymatic	Cyclic Voltammetry	Glucose	1–8 mM	Not reported	[208]
*Nafion-Magnetite-CNT hybrid nanocomposite*	Enzymatic	Amperometry	Hydrogen Peroxide	1.2–21.6 μM	3.7 nM	[209]
*Fe_2_O_3_ nanoparticles + MWCNTs + Chitosan*	Non-enzymatic	Differential Pulse Voltammetry	Levodopa	0.3–8 μM	0.24 μM	[210]
*Graphene oxide loaded maghemite superparamagnetic nanoparticles*	Non-enzymatic	Cyclic Voltammetry	miRNA-21	Not reported	1 fM	[211]
*Magnetite@SiO_2_@Au core-shell nanoparticles coated with cDNA*	Non-enzymatic (HCR)	Impedance Spectroscopy	miRNA-126	5–5000 fM	2 fM	[196]
*Nanostructured Fe_2_O_3_ + PEDOT: PSS*	Immuno-sensing	Amperometry	CEA	4–25 ng/mL	Not reported	[197]
*Prussian blue/Graphene oxide + spiky Au@magnetite nanoparticles*	Non-enzymatic	Cyclic Voltammetry	MCF-7 exosomes	200—5 × 10^5^ particles/μL	80 particles/μL	[198]
*Magnetite + covalent organic framework + Au nanoparticles*	Aptameric	Cyclic Voltammetry and Impedance Spectroscopy	ATP	5 pM–50 μM	1.6 pM	[199]
*Maghemite nanoparticles (different morphologies)*	Non-enzymatic	Amperometry	Dopamine Uric Acid	Dopamine: 0.15–75 μM Uric Acid: 5 μM–0.15 mM	Dopamine: 0.15 μM Uric Acid: 5 μM	[200]
*Fe_2_O_3_ nanoparticles*	Non-enzymatic	Differential Pulse Voltammetry	Epinephrine	0.05–15 μM	1.6 μM	[202]

## Data Availability

Not applicable.

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
