# Peer review of "Iron Oxide Nanoparticles: A Review on the Province of Its Compounds, Properties and Biological Applications"

_materials, 2022, doi:10.3390/ma16010059_

Round 1

Reviewer 1 Report

This review is a detailed and systematic work. It summarizes the development of iron oxide in recent years and looks forward to it. I strongly recommend accepting it.

Author Response

We deeply thank the reviewer for their kind words, appreciation, and consideration of publication of our work.

Reviewer 2 Report

The authors made a huge work. The prepared manuscript is very interesting. The comprehensive and thorough review of all known iron nanoparticles was given. I really enjoyed reading it. Nevertheless, there can be do several improvements. The references, especially in subsection 6.4. Iron oxide nanoparticles in biosensing applications are quite old. I have found this part as the worst written part of whole manuscript. I am aware there are numerous electroanalytical techniques and methods, but the authors had to focus on a five-year period. They must redo subsection 6.4, at least by expanding with the newer references. On the other hand, I found several wrong symbols, A° instead of Å (Swedish scientist Anders Jonas Ångström) in Lines 213, 313, 314, 428, 484, 485 and 610; in Lines 325 and 738 "900" instead "90°" and similar in Line 1394 "0C" instead of "°C"; Lines 465, 482 and 595 need a space between subsection number and a title.  

Author Response

We deeply thank the reviewer for their kind words and acknowledgement of our extensive review work.

Subsection 6.4 has been reworked and expanded to include recent and diverse electroanalytical techniques, methods and strategies employed over the last five years involving the use of iron oxide nanoparticles, as per reviewer’s comments.
With regards to the wrong symbols, they have been duly corrected at all the mentioned instances and have been checked through the entirety of the revised manuscript for correctness.

Reviewer 3 Report

This is a well written and very extensive review. However, I do have some serious concerns. The present illustrations are less useful and almost all are reproductions from earlier publications. It is essential to include figures relevant for the text. I would like to see structures of e.g. the different materials, overviews of how the chemical modifications are made etc. More tables would also facilite the reading of the manuscript. The manuscript is simply too ”heavy” in its present form.

Author Response

We deeply thank the reviewer for their kind words about our extensive review. As per the reviewer’s suggestions, new illustrations on overview of the review, crystal structures of different iron oxide polymorphs (Fig 1, Fig 2 in revised manuscript) have been included. An additional table summarizing the electroanalytical methods have been included as well (Table 2 in revised manuscript).

Reviewer 4 Report

Manuscript Number: materials-2035556; Title: Iron Oxide Nanoparticles: A Review on The Province of Its Compounds, Properties And Biological Applications.

This review is a good fit for the journal Materials. Iron Oxide nanoparticles are an important class of materials, and their applications in various multidimensional areas are extensively demonstrated. However, this review is unique and comprehensively described the various Iron nanoparticles, their synthesis protocols, and future perspectives of the various applications domains of iron oxide nanoparticles. The diverse properties of iron oxide nanoparticles employed in the fields of imaging, sensing, diagnostics, and therapeutics, have paved the way to more efficient disease diagnosis and cure with additional benefits of very minimal side effects were also elaborately discussed in this review.  Many sections in this paper are nicely written with highly scientific information. The body of the literature reviewed is typically published in recent journals, and a systematic literature survey has been done by the authors. The introduction is well structured and it indicates the purpose of this review; the review needs additional points to enhance the impact of this work. There are some comments that need to be considered before acceptance of this paper.

Comments

1.    Introduction: Authors need to include some of the schematic presentations of the overview of the current review on Iron Oxide Nanoparticles with their potential applications to capture the attention of the readers. 

2.    Provide a paragraph on the “green" synthesis of iron oxide nanoparticles and their stabilization properties.

3.    One paragraph should deal with the environmental remediation of Iron oxide nanoparticles and their potential contribution towards water purification.

4.    It would be nice to provide SEM or TEM images of iron oxide nanoparticles and some of the mechanisms of the stabilization of iron oxide nanoparticles using various surfactants or polymers.

Author Response

We deeply thank the reviewer for their kind words and appreciation about our review. Here is our response to each of their points.

  1. Introduction: A schematic on the overview of the current review has been prepared and included as Fig 1 in the revised manuscript as suggested to capture the attention of the readers
  2. A separate section on “green” synthesis of iron oxide nanoparticles (section 2.7) has been included in section 2 as suggested with discussions on their stabilization properties
  3. Our review’s objective was to cover the broad spectrum of biological applications of iron oxide nanoparticles. We are not quite sure about how addition of a paragraph on environmental remediation of iron oxide nanoparticles would fit in this regard
  4. As suggested by the reviewer, SEM and TEM images of some of the iron oxide nanoparticles with their stabilization mechanisms have been included in the revised manuscript (Fig 3 and 7)

Round 2

Reviewer 3 Report

The authors have been modified the manuscript and it can be published in its present form.